# HYBRID RL: USING BOTH OFFLINE AND ONLINE DATA CAN MAKE RL EFFICIENT

**Yuda Song**[*]
Carnegie Mellon University

**Yifei Zhou**[*]
Cornell University

**Ayush Sekhari**
MIT

**J. Andrew Bagnell**
Carnegie Mellon University

**Akshay Krishnamurthy**
Microsoft Research

**Wen Sun**
Cornell University

## ABSTRACT

We consider a hybrid reinforcement learning setting (Hybrid RL), in which an agent has access to an offline dataset and the ability to collect experience via real-world online interaction. The framework mitigates the challenges that arise in both pure offline and online RL settings, allowing for the design of simple and highly effective algorithms, in both theory and practice. We demonstrate these advantages by adapting the classical Q learning/iteration algorithm to the hybrid setting, which we call Hybrid Q-Learning or Hy-Q. In our theoretical results, we prove that the algorithm is both computationally and statistically efficient whenever the offline dataset supports a high-quality policy and the environment has bounded bilinear rank. Notably, we require no assumptions on the coverage provided by the initial distribution, in contrast with guarantees for policy gradient/iteration methods. In our experimental results, we show that Hy-Q with neural network function approximation outperforms state-of-the-art online, offline, and hybrid RL baselines on challenging benchmarks, including Montezuma's Revenge.

## 1 INTRODUCTION

Learning by interacting with an environment, in the standard online reinforcement learning (RL) protocol, has led to impressive results across a number of domains. State-of-the-art RL algorithms are quite general, employing function approximation to scale to complex environments with minimal domain expertise and inductive bias. However, online RL agents are also notoriously sample inefficient, often requiring billions of environment interactions to achieve suitable performance. This issue is particularly salient when the environment requires sophisticated exploration and a high quality reset distribution is unavailable to help overcome the exploration challenge. As a consequence, the practical success of online RL and related policy gradient/improvement methods has been largely restricted to settings where a high quality simulator is available.

To overcome the issue of sample inefficiency, attention has turned to the offline RL setting (Levine et al., 2020), where, rather than interacting with the environment, the agent trains on a large dataset of experience collected in some other manner (e.g., by a system running in production or an expert). While these methods still require a large dataset, they mitigate the sample complexity concerns of online RL, since the dataset can be collected without compromising system performance. However, offline RL methods can suffer from *distribution shift*, where the state distribution induced by the learned policy differs significantly from the offline distribution (Wang et al., 2021). Existing provable approaches for addressing distribution shift are computationally intractable, while empirical approaches rely on heuristics that can be sensitive to the domain and offline dataset (as we will see).

In this paper, we focus on a hybrid reinforcement learning setting, which we call Hybrid RL, that draws on the favorable properties of both offline and online settings. In Hybrid RL, the agent has both an offline dataset and the ability to interact with the environment, as in the traditional online RL setting. The offline dataset helps address the exploration challenge, allowing us to greatly reduce

---

[*]Equal contribution
Author contact info: `yudas@andrew.cmu.edu`, `yz639@cornell.edu`, `sekhari@mit.edu`, `dbagnell@aurora.tech`, `akshaykr@microsoft.com`, `ws455@cornell.edu`

the number of interactions required. Simultaneously, we can identify and correct distribution shift issues via online interaction. Variants of the setting have been studied in a number of empirical works (Rajeswaran et al., 2017; Hester et al., 2018; Nair et al., 2018; 2020; Vecerik et al., 2017) which mainly focus on using expert demonstrations as offline data. Our algorithmic development is closely related to these works, although our focus is on formalizing the hybrid setting and establishing theoretical guarantees against more general offline datasets.

Hybrid RL is closely related to the *reset setting*, where the agent can interact with the environment starting from a "nice" distribution. A number of simple and effective algorithms, including CPI (Kakade & Langford, 2002), PSDP (Bagnell et al., 2003), and policy gradient methods (Kakade, 2001; Agarwal et al., 2020b)—which have further inspired deep RL methods such as TRPO (Schulman et al., 2015) and PPO (Schulman et al., 2017)—are provably efficient in the reset setting. Yet, a nice reset distribution is a strong requirement (often tantamount to having access to a detailed simulation) and unlikely to be available in real world applications. Hybrid RL differs from the reset setting in that (a) we have an offline dataset, but (b) our online interactions start from the initial distribution of the environment, which is not assumed to have any nice properties. Both features (offline data and a nice reset distribution) facilitate algorithm design by de-emphasizing the exploration challenge. However, Hybrid RL is much more practical since an offline dataset is much easier to obtain in practice.

We showcase the Hybrid RL setting with a new algorithm, Hybrid Q learning or Hy-Q (pronounced: Haiku). The algorithm is a simple adaptation of the classical fitted Q-iteration algorithm (FQI) and accommodates value-based function approximation.[1] For our theoretical results, we prove that Hy-Q is both statistically and computationally efficient assuming that: (1) the offline distribution covers some high quality policy, (2) the MDP has low bilinear rank, (3) the function approximator is Bellman complete, and (4) we have a least squares regression oracle. The first three assumptions are standard statistical assumptions in the RL literature while the fourth is a widely used computational abstraction for supervised learning. No computationally efficient algorithms are known under these assumptions in pure offline or pure online settings, which highlights the advantages of the hybrid setting.

We also implement Hy-Q and evaluate it on two challenging RL benchmarks: a rich observation combination lock (Misra et al., 2020) and Montezuma's Revenge from the Arcade Learning Environment (Bellemare et al., 2013). Starting with an offline dataset that contains some transitions from a high quality policy, our approach outperforms: an online RL baseline with theoretical guarantees, an online deep RL baseline tuned for Montezuma's Revenge, pure offline RL baselines, imitation learning baselines, and existing hybrid methods. Compared to the online methods, Hy-Q requires only a small fraction of the online experience, demonstrating its sample efficiency. Compared to the offline and hybrid methods, Hy-Q performs most favorably when the offline dataset also contains many interactions from low quality policies, demonstrating its robustness. These results reveal the significant benefits that can be realized by combining offline and online data.

## 2 RELATED WORKS

We discuss related works from four categories: pure online RL, online RL with access to a reset distribution, offline RL, and prior work in hybrid settings. We note that pure online RL refers to the setting where one can only reset the system to initial state distribution of the environment, which is not assumed to provide any form of coverage.

**Pure online RL** Beyond tabular settings, many existing statistically efficient RL algorithms are not computationally tractable, due to the difficulty of implementing optimism. This is true in the linear MDP (Jin et al., 2020) with large action spaces, the linear Bellman complete model (Zanette et al., 2020; Agarwal et al., 2019), and in the general function approximation setting (Jiang et al., 2017; Sun et al., 2019; Du et al., 2021; Jin et al., 2021a). These computational challenges have inspired results on intractability of aspects of online RL (Dann et al., 2018; Kane et al., 2022).

There are several online RL algorithms that aim to tackle the computational issue via stronger structural assumptions and supervised learning-style computational oracles (Misra et al., 2020; Zhang et al., 2022c; Agarwal et al., 2020a; Uehara et al., 2021; Modi et al., 2021; Zhang et al., 2022a; Qiu et al., 2022). Compared to these oracle-based methods, our approach operates in the more general

---

[1]We use Q-learning and Q-iteration interchangeably, although they are not strictly speaking the same algorithm. Our theoretical results analyze Q-iteration, but we use an algorithm with an online/mini-batch flavor that is closer to Q-learning for our experiments.

"bilinear rank" setting and relies on a standard supervised learning primitive: least squares regression. Notably, our oracle admits efficient implementation with linear function approximation, so we obtain an end-to-end computational guarantee; this is not true for prior oracle-based methods.

There are many deep RL methods for the online setting (e.g., Schulman et al. (2015; 2017); Lillicrap et al. (2016); Haarnoja et al. (2018); Schrittwieser et al. (2020)). Apart from a few exceptions (e.g., Burda et al. (2018); Badia et al. (2020); Guo et al. (2022)), most rely on random exploration and are not capable of strategic exploration. In our experiments, we test our approach on Montezuma's Revenge, and we pick RND (Burda et al., 2018) as a deep RL exploration baseline due to its effectiveness.

**Online RL with reset distributions**  When an exploratory reset distribution is available, a number of statistically and computationally efficient algorithms are known. The classic algorithms are CPI (Kakade & Langford, 2002), PSDP (Bagnell et al., 2003), Natural Policy Gradient (Kakade, 2001; Agarwal et al., 2020b), and POLYTEX (Abbasi-Yadkori et al., 2019). Uchendu et al. (2022) recently demonstrated that algorithms like PSDP work well when equipped with modern neural network function approximators. However, these algorithms (and their analyses) heavily rely on the reset distribution to mitigate the exploration challenge, but such a reset distribution is typically unavailable in practice, unless one also has a simulator. In contrast, we assume the offline data covers some high quality policy, which helps with exploration, but we do not require an exploratory reset distribution. This makes the hybrid setting much more practically appealing.

**Offline RL**  Offline RL methods learn policies solely from a given offline dataset, with no interaction whatsoever. When the dataset has global coverage, algorithms such as FQI (Munos & Szepesvári, 2008; Chen & Jiang, 2019) or certainty-equivalence model learning (Ross & Bagnell, 2012), can find near-optimal policies in an oracle-efficient manner, via least squares or model-fitting oracles. However, with only partial coverage, existing methods either (a) are not computationally efficient due to the difficulty of implementing pessimism both in linear settings with large action spaces (Jin et al., 2021b; Zhang et al., 2022b; Chang et al., 2021) and general function approximation settings (Uehara & Sun, 2021; Xie et al., 2021a; Jiang & Huang, 2020; Chen & Jiang, 2022; Zhan et al., 2022), or (b) require strong representation conditions such as policy-based Bellman completeness (Xie et al., 2021a; Zanette et al., 2021). In contrast, in the hybrid setting, we obtain an efficient algorithm under the more natural condition of completeness w.r.t., the Bellman optimality operator only.

Among the many empirical offline RL methods (e.g., Kumar et al. (2020); Yu et al. (2021); Kostrikov et al. (2021); Fujimoto & Gu (2021)), we use CQL (Kumar et al., 2020) as a baseline in our experiments, since it has been shown to work in image-based control settings such as Atari games.

**Online RL with offline datasets**  Ross & Bagnell (2012) developed a model-based algorithm for a similar hybrid setting. In comparison, our approach is model-free and consequently more suitable for high-dimensional state spaces (e.g., raw-pixel images). Xie et al. (2021b) studied hybrid RL and show that offline data does not yield statistical improvements in tabular MDPs. Our work instead focuses on the function approximation setting and demonstrates computational benefits of hybrid RL.

On the empirical side, several works consider combining offline expert demonstrations with online interaction (Rajeswaran et al., 2017; Hester et al., 2018; Nair et al., 2018; 2020; Vecerik et al., 2017). A common challenge in offline RL is the robustness against low-quality offline dataset. Previous works mostly focus on expert demonstrations and have no rigorous guarantees for such robustness. In fact, Nair et al. (2020) showed that such degradation in performance indeed happens in practice with low-quality offline data. In our experiments, we observe that DQfD (Hester et al., 2018) also has a similar degradation. On the other hand, our algorithm is robust to the quality of the offline data. Note that the core idea of our algorithm is similar to that of Vecerik et al. (2017), who adapt DDPG to the setting of combining RL with expert demonstrations for continuous control. Although Vecerik et al. (2017) does not provide any theoretical results, it may be possible to combine our theoretical insights with existing analyses for policy gradient methods to establish some guarantees of the algorithm from Vecerik et al. (2017) for the hybrid RL setting. We also include a detailed comparison with previous empirical work in Appendix D.

## 3  PRELIMINARIES

We consider finite horizon Markov Decision Process $M(\mathcal{S}, \mathcal{A}, H, R, P, d_0)$, where $\mathcal{S}$ is the state space, $\mathcal{A}$ is the action space, $H$ denotes the horizon, stochastic rewards $R(s, a) \in \Delta([0, 1])$ and $P(s, a) \in \Delta(\mathcal{S})$ are the reward and transition distributions at $(s, a)$, and $d_0 \in \Delta(\mathcal{S})$ is the initial

---

**Algorithm 1** Hybrid Q-learning using both offline and online data (Hy-Q)

---

**Require:** Value class: $\mathcal{F}$, #iterations: $T$, offline dataset $\mathcal{D}_h^\nu$ of size $m_{\text{off}} = T$ for $h \in [H-1]$.
1: Initialize $f_h^1(s, a) = 0$.
2: **for** $t = 1, \ldots, T$ **do**
3: Let $\pi^t$ be the greedy policy w.r.t. $f^t$ i.e., $\pi_h^t(s) = \text{argmax}_a f_h^t(s, a)$.
4: For each $h$, collect $m_{\text{on}} = 1$ online tuples $\mathcal{D}_h^t \sim d_h^{\pi^t}$. **// Online collection**
  **// FQI using both online and offline data**
5: Set $f_H^{t+1}(s, a) = 0$.
6: **for** $h = H-1, \ldots, 0$ **do**
7:  Estimate $f_h^{t+1}$ using least squares regression on the aggregated data $\mathcal{D}_h^t = \mathcal{D}_h^\nu + \sum_{\tau=1}^t \mathcal{D}_h^\tau$:

$$f_h^{t+1} \leftarrow \underset{f \in \mathcal{F}_h}{\text{argmin}}\left\{ \widehat{\mathbb{E}}_{\mathcal{D}_h^t} (f(s, a) - r - \max_{a'} f_{h+1}^{t+1}(s', a'))^2 \right\} \tag{1}$$

8: **end for**
9: **end for**

---

distribution. We assume the agent can only reset from $d_0$ (at the beginning of each episode). Since the optimal policy is non-stationary in this setting, we define a policy $\pi := \{\pi_0, \ldots, \pi_{H-1}\}$ where $\pi_h : \mathcal{S} \mapsto \Delta(\mathcal{A})$. Given $\pi$, $d_h^\pi \in \Delta(\mathcal{S} \times \mathcal{A})$ denotes the state-action occupancy induced by $\pi$ at step $h$.

Given $\pi$, we define the state and state-action value functions in the usual manner: $V_h^\pi(s) = \mathbb{E}[\sum_{\tau=h}^{H-1} r_\tau | \pi, s_h = s]$ and $Q_h^\pi(s, a) = \mathbb{E}[\sum_{\tau=h}^{H-1} r_\tau | \pi, s_h = s, a_h = a]$. $Q^\star$ and $V^\star$ denote the optimal value functions. We define the Bellman operator $\mathcal{T}$ such that for any $f : \mathcal{S} \times \mathcal{A} \mapsto \mathbb{R}$,

$$\mathcal{T}f(s, a) = \mathbb{E}[R(s, a)] + \mathbb{E}_{s' \sim P(s,a)} \max_{a'} f(s', a') \qquad \forall s, a,$$

We assume that for each $h$ we have an offline dataset of $m_{\text{off}}$ samples $(s, a, r, s')$ drawn iid via $(s, a) \sim \nu_h, r \in R(s, a), s' \sim P(s, a)$. Here $\nu = \{\nu_0, \ldots, \nu_{H-1}\}$ denote the corresponding offline data distributions. For a dataset $\mathcal{D}$, we use $\widehat{\mathbb{E}}_{\mathcal{D}}[\cdot]$ to denote a sample average over this dataset. For our theoretical results, we will assume that $\nu$ covers some high-quality policy.

We consider the value-based function approximation setting, where we are given a function class $\mathcal{F} = \mathcal{F}_0 \times \cdots \times \mathcal{F}_{H-1}$ with $\mathcal{F}_h \subset \mathcal{S} \times \mathcal{A} \mapsto [0, V_{\max}]$ that we use to approximate the value functions for the underlying MDP. For ease of notation, we define $f = \{f_0, \ldots, f_{H-1}\}$ and define $\pi^f$ to be the greedy policy w.r.t., $f$, which chooses actions as $\pi_h^f(s) = \text{argmax}_a f_h(s, a)$.

## 4 HYBRID Q-LEARNING

In this section, we present our algorithm *Hybrid Q Learning* – Hy-Q in Algorithm 1. Hy-Q takes an offline dataset $\mathcal{D}^\nu$ that contains $(s, a, r, s')$ tuples and a Q function class $\mathcal{F} \subset \mathcal{S} \times \mathcal{A} \mapsto [0, H]$ as inputs, and outputs a policy that optimizes the given reward function. The algorithm is conceptually simple: it iteratively executes the Fitted Q Iteration procedure (line 6) using the offline dataset *and* on-policy samples generated by the learned policies.

Specifically, at iteration $t$, we have an estimate $f^t$ of the $Q^\star$ function and we set $\pi^t$ to be the greedy policy for $f^t$. We execute $\pi^t$ to collect a dataset $\mathcal{D}_h^t$ of online samples in line 4. Then we run FQI, a dynamic programming style algorithm on both the offline dataset $\mathcal{D}^\nu$ and all previously collected online samples $\{\mathcal{D}_h^\tau\}_{\tau=1}^t$. The FQI update works backward from time step $H$ to 0 and computes $f_h^{t+1}$ via least squares regression with input $(s, a)$ and regression target $r + \max_{a'} f_{h+1}^{t+1}(s', a')$.[2]

Let us make several remarks. Intuitively, the FQI updates in Hy-Q try to ensure that the estimate $f^t$ has small Bellman error under both the offline distribution $\nu$ and the online distributions $d^{\pi^t}$. The standard offline version of FQI ensures the former, but this alone is insufficient when the offline dataset has poor coverage. Indeed FQI may have poor performance in such cases (see examples in Zhan et al., 2022; Chen & Jiang, 2022). The key insight in Hy-Q is to use online interaction to ensure that we also have small Bellman error on $d^{\pi^t}$. As we will see, the moment we find an $f^t$ that has small Bellman error on the offline distribution $\nu$ and *its own greedy policy's distribution* $d^{\pi^t}$, FQI

---

[2]Note that FQI and Hy-Q extend to the infinite horizon discounted setting (Munos & Szepesvári, 2008).

guarantees that $\pi^t$ will be at least as good as *any* policy covered by $\nu$. This observation results in an explore-or-terminate phenomenon: either $f^t$ has small Bellman error on its distribution and we are done, or $d^{\pi^t}$ must be significantly different from distributions we have seen previously and we make progress. Crucially, no explicit exploration is required for this argument, which is precisely how we avoid the computational difficulties with implementing optimism.

Another important point pertains to *catastrophic forgetting*. We will see that the size of the offline dataset $m_{\text{off}}$ should be comparable to the total amount of online data $\{\mathcal{D}_h^\tau\}_{\tau=1}^T$, so that the two terms in Eq. 1 have similar weight and we ensure low Bellman error on $\nu$ throughout the learning process. In practice, we implement this by having all model updates use a fixed proportion of offline samples even as we collect more online data, so that we do not "forget" the distribution $\nu$. This is quite different from warm-starting with $\mathcal{D}^\nu$ and then switching to online RL, which may result in catastrophic forgetting due to a vanishing proportion of offline samples being used for model training as we collect more online samples. We note that this balancing scheme is analogous to and inspired by the one used by Ross & Bagnell (2012) in the context of model-based RL with a reset distribution. Previously, similar techniques have also been explored for various applications (for example, see Appendix F.3 of Kalashnikov et al. (2018)). As in Ross & Bagnell (2012), a key practical insight from our analysis is that the offline data should be used throughout training to avoid catastrophic forgetting.

## 5 THEORETICAL ANALYSIS: LOW BILINEAR RANK MODELS

In this section we present the main theoretical guarantees for Hy-Q. We start by stating the main assumptions and definitions for the function approximator, the offline data distribution, and the MDP. We state the key definitions and then provide some discussion.

**Assumption 1** (Realizability and Bellman completeness). *For any $h$, we have $Q_h^\star \in \mathcal{F}_h$. Additionally, for any $f_{h+1} \in \mathcal{F}_{h+1}$, we have $\mathcal{T}f_{h+1} \in \mathcal{F}_h$.*

**Definition 1** (Bellman error transfer coefficient). *For any policy $\pi$, define the transfer coefficient as*

$$C_\pi := \max\left\{ 0, \ \max_{f \in \mathcal{F}} \frac{\sum_{h=0}^{H-1} \mathbb{E}_{s,a \sim d_h^\pi}\left[\mathcal{T}f_{h+1}(s,a) - f_h(s,a)\right]}{\sqrt{\sum_{h=0}^{H-1} \mathbb{E}_{s,a \sim \nu_h}\left(\mathcal{T}f_{h+1}(s,a) - f_h(s,a)\right)^2}} \right\}. \tag{2}$$

**Definition 2** (Bilinear model (Du et al., 2021)). *We say that the MDP together with the function class $\mathcal{F}$ is a bilinear model of rank $d$ if for any $h \in [H-1]$, there exist two (unknown) mappings $X_h, W_h : \mathcal{F} \mapsto \mathbb{R}^d$ with $\max_f \|X_h(f)\|_2 \leq B_X$ and $\max_f \|W_h(f)\|_2 \leq B_W$ such that:*

$$\forall f, g \in \mathcal{F}: \ \left|\mathbb{E}_{s,a \sim d_h^{\pi_f}}\left[g_h(s,a) - \mathcal{T}g_{h+1}(s,a)\right]\right| = |\langle X_h(f), W_h(g)\rangle|.$$

All concepts defined above are frequently used in the statistical analysis of RL methods with function approximation. Realizability is the most basic function approximation assumption, but is known to be insufficient for offline RL (Foster et al., 2021) unless other strong assumptions hold (Xie & Jiang, 2021; Zhan et al., 2022; Chen & Jiang, 2022). Completeness is the most standard strengthening of realizability that is used routinely in both online (Jin et al., 2021a) and offline RL (Munos & Szepesvári, 2008; Chen & Jiang, 2019) and is known to hold in several settings including the linear MDP and the linear quadratic regulator. These assumptions ensure that the dynamic programming updates of FQI are stable in the presence of function approximation.

The transfer coefficient definition above is somewhat non-standard, but is actually weaker than related notions used in prior offline RL results. First, the average Bellman error appearing in the numerator is weaker than the squared Bellman error notion of (Xie et al., 2021a); a simple calculation shows that $C_\pi^2$ is upper bounded by their coefficient. Second, by using Bellman errors, both of these are bounded by notions involving density ratios (Kakade & Langford, 2002; Munos & Szepesvári, 2008; Chen & Jiang, 2019). Finally, many works, particularly those that do not employ pessimism (Munos & Szepesvári, 2008; Chen & Jiang, 2019), require "all-policy" analogs, which places a much stronger requirement on the offline data distribution $\nu$. In contrast, we will only ask that $C_\pi$ is small for *some* high-quality policy that we hope to compete with (see Appendix A.5 for more details).

Lastly, the bilinear model was developed in a series of works (Jiang et al., 2017; Jin et al., 2021a; Du et al., 2021) on sample efficient online RL.[3] The setting is known to capture a wide class of models

---

[3] Jin et al. (2021a) consider the Bellman Eluder dimension, which is related but distinct from the Bilinear model. However, our proofs can be easily translated to this setting; see Appendix C for more details.

including linear MDPs, linear Bellman complete models, low-rank MDPs, reactive POMDPs, and more. As a technical note, the main paper focuses on the "Q-type" version of the bilinear model, but the algorithm and proofs easily extend to the "V-type" version. See Appendix C for details.

**Theorem 1** (Cumulative suboptimality). *Fix $\delta \in (0, 1)$, $m_{\text{off}} = T$ and $m_{\text{on}} = 1$, suppose that the function class $\mathcal{F}$ satisfies Assumption 1, and together with the underlying MDP admits Bilinear rank $d$. Then with probability at least $1 - \delta$, Algorithm 1 obtains the following bound on cumulative subpotimality w.r.t. any comparator policy $\pi^e$,*

$$\sum_{t=1}^{T} V^{\pi^e} - V^{\pi^t} = \widetilde{O}\Big(\max\{C_{\pi^e}, 1\} V_{\max} \sqrt{dH^2 T \cdot \log(|\mathcal{F}|/\delta)}\Big),$$

*where $\pi^t = \pi^{f^t}$ is the greedy policy w.r.t. $f^t$ at round $t$.*

A standard online-to-batch conversion (Shalev-Shwartz & Ben-David, 2014) immediately gives the following sample complexity guarantee for Algorithm 1 for finding an $\epsilon$-suboptimal policy w.r.t. the optimal policy $\pi^*$ for the underlying MDP.

**Corollary 1** (Sample complexity). *Under the assumptions of Theorem 1 if $C_{\pi^*} < \infty$ then Algorithm 1 can find an $\epsilon$-suboptimal policy $\widehat{\pi}$ for which $V^{\pi^*} - V^{\widehat{\pi}} \leq \epsilon$ with total sample complexity:*

$$n = \widetilde{O}\big(V_{\max}^2 C_{\pi^*}^2 H^3 d \log(|\mathcal{F}|/\delta)/\epsilon^2\big)$$

The results formalize the statistical properties of Hy-Q. In terms of sample complexity, a somewhat unique feature of the hybrid setting is that both transfer coefficient and bilinear rank parameters are relevant, whereas these (or related) parameters typically appear in isolation in offline and online RL respectively. In terms of coverage, Theorem 1 highlights an "oracle property" of Hy-Q: it competes with *any* policy that is sufficiently covered by the offline dataset.

We also highlight the computational efficiency of Hy-Q: it only requires solving least squares problems over the function class $\mathcal{F}$. To our knowledge, no purely online or purely offline methods are known to be efficient in this sense, except under much stronger "uniform" coverage conditions.

### 5.1 THE LINEAR BELLMAN COMPLETENESS MODEL

We next showcase one example of low bilinear rank models: the popular linear Bellman complete model which captures the linear MDP model (Yang & Wang, 2019; Jin et al., 2020), and instantiate the sample complexity bound in Corollary 1.

**Definition 3.** *Given a feature function $\phi : \mathcal{S} \times \mathcal{A} \mapsto \mathbb{B}_d(1)$, a model admits linear Bellman completeness if for any $w \in \mathbb{B}_d(B_W)$, there exists a $w' \in \mathbb{B}_d(B_W)$ such that*

$$\forall s, a : \qquad \langle w', \phi(s, a) \rangle = \mathbb{E}[R(s, a)] + \mathbb{E}_{s' \sim P(s,a)} \max_{a'} \langle w, \phi(s', a') \rangle.$$

Note that the above condition implies that $Q_h^\star(s, a) = \langle w_h^\star, \phi(s, a) \rangle$ with $\|w_h^\star\|_2 \leq B_W$. Thus, we can define a function class $\mathcal{F}_h = \{\langle w_h, \phi(s, a) \rangle : w_h \in \mathbb{R}^d, \|w_h\|_2 \leq B_W\}$ which by inspection satisfies Assumption 1. Additionally, this model is also known to have bilinear rank at most $d$ (Du et al., 2021). Thus, using Corollary 1 we immediately get the following guarantee:

**Lemma 1.** *Let $\delta \in (0, 1)$, suppose the MDP is linear Bellman complete, $C_{\pi^*} < \infty$, and consider $\mathcal{F}_h$ defined above. Then, with probability $1 - \delta$, Algorithm 1 finds an $\epsilon$-suboptimal policy with total sample complexity:*

$$n = \widetilde{O}\big(B_W^2 C_{\pi^*}^2 H^4 d^2 \log(1/\delta)/\epsilon^2\big).$$

**Remark 1** (Computational efficiency). *For linear Bellman complete models, we note that Algorithm 1 can be implemented efficiently under mild assumptions. For the class $\mathcal{F}$ in Lemma 1, the regression problem in (1) reduces to a least squares linear regression with a norm constraint on the weight vector. This regression problem can be solved efficiently by convex programming with computational efficiency scaling polynomially in the number of parameters (Bubeck et al., 2015) (d here), whenever $\max_a f_{h+1}(s, a)$ (or $\operatorname{argmax}_a f_{h+1}(s, a)$) can be computed efficiently.*

**Remark 2.** *(Linear MDPs) Since linear Bellman complete models generalize linear MDPs (Yang & Wang, 2019; Jin et al., 2020), as we discuss above, Algorithm 1 can be implemented efficiently whenever $\max_a f_{h+1}(s, a)$ can be computed efficiently. The latter is tractable when:*

- *When $|\mathcal{A}|$ is small/finite, one can just enumerate to compute $\max_a f_{h+1}(s, a)$ for any $s$, and thus (1) can be implemented efficiently. The computational efficiently of Algorithm 1 in this case is comparable to the prior works, e.g. Jin et al. (2020).*

- *When the set $\{\phi(s, a) \mid a \in \mathcal{A}\}$ is convex and compact, one can simply use a linear optimization oracle to compute $\max_a f_{h+1}(s, a) = \max_a w_{h+1}^\top \phi(s, a)$. This linear optimization problem is itself solvable with computational efficiency scaling polynomially with $d$. here).*

  *Note that even under access to a linear optimization oracle, prior works e.g. Jin et al. (2020) rely on bonuses in the form of $\operatorname{argmax}_a \phi(s, a)^\top w + \beta \sqrt{\phi(s, a)^\top \Sigma \phi(s, a)}$, where $\Sigma$ is some positive definite matrix (e.g., the regularized feature covariance matrix). Computing such bonuses could be NP-hard (in the feature dimension $d$) without additional assumptions (Dani et al., 2008).*

**Remark 3.** *(Relative condition number) A common coverage metric in these linear MDP models is the relative condition number. In Appendix A.5, we show that our coefficient $C_\pi$ is upper bounded by the relative condition number of $\pi$ with respect to $\nu$: $\mathbb{E}_{d^\pi} \|\phi\|_{\Sigma_\nu^{-1}}$, where $\Sigma_\nu = \mathbb{E}_{s,a \sim \nu} \phi(s, a)\phi^\top(s, a)$. Concretely, we have $C_\pi \leq \sqrt{\max_h \mathbb{E}_{d_h^\pi} \|\phi\|_{\Sigma_{\nu_h}^{-1}}^2}$.*

## 5.2 WHY DON'T OFFLINE RL METHODS WORK?

One may wonder why do pure offline RL methods fail to learn when the transfer coefficient is bounded, and why does online access help? We illustrate with the MDP construction developed by Zhan et al. (2022); Chen & Jiang (2022), visualized in Figure 1.

Consider two MDPs $\{M_1, M_2\}$ with $H = 2$, three states $\{A, B, C\}$, two actions $\{L, R\}$ and the fixed start state $A$. The two MDPs have the same dynamics but different rewards. In both, actions from state $B$ yield reward 1. In $M_1$, $(C, R)$ yields reward 1 while $(C, L)$ yields reward 1 in $M_2$. All other rewards are 0. In both $M_1$ and $M_2$, an optimal policy is $\pi^*(A) = L$ and $\pi^*(B) = \pi^*(C) = \mathrm{Uniform}(\{L, R\})$. With $\mathcal{F} = \{Q_1^\star, Q_2^\star\}$ where $Q_j^\star$ is the optimal $Q$ function for $M_j$, then one can easily verify that $\mathcal{F}$ satisfies Bellman completeness, for both MDPs. Finally with offline distribution $\nu$ supported on states $A$ and $B$ only (with no coverage on state $C$), we have sufficient coverage over $d^{\pi^*}$. However, samples from $\nu$ are unable to distinguish between $f_1$ and $f_2$ or ($M_1$ and $M_2$), since state $C$ is not supported by $\nu$. Unfortunately, adversarial tie-breaking may result the greedy policies of $f_1$ and $f_2$ visiting state $C$, where we have no information about the correct action.

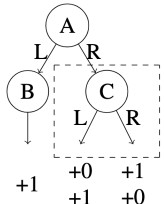

Figure 1: A hard instance for offline RL (Zhan et al., 2022, reproduced with permission)

This issue has been documented before, and in order to address it with pure offline RL, existing approaches require additional structural assumptions. For instance, Chen & Jiang (2022) assume that $Q^\star$ has a gap, which usually does not hold when action space is large or continuous. Xie et al. (2021a) assumes policy-dependent Bellman completeness for every possible policy $\pi \in \Pi$ (which is much stronger than our assumption), and Zhan et al. (2022) assumes a somewhat non-interpretable realizability assumption on some "value" function that does not obey the standard Bellman equation. In contrast, by combining offline data and online data, our approach focuses on functions that have small Bellman residual under both the offline distribution and the on-policy distributions, which together with the offline data coverage assumption, ensures near optimality. It is easy to see that the hybrid approach will succeed Figure 1.

## 6 EXPERIMENTS

In this section we discuss empirical results comparing Hy-Q to several representative RL methods on two challenging benchmarks. Our experiments focus on answering the following questions:

1. Can Hy-Q efficiently solve problems that SOTA offline RL methods simply cannot?

2. Can Hy-Q, via the use of offline data, significantly improve the sample efficiency of online RL?

3. Does Hy-Q scale to challenging deep-RL benchmarks?

Our empirical results provide positive answers to all of these questions. To study the first two, we consider the diabolical combination lock environment (Misra et al., 2020; Zhang et al., 2022c), a synthetic environment designed to be particularly challenging for online exploration. The synthetic nature allows us to carefully control the offline data distribution to modulate the difficulty of the

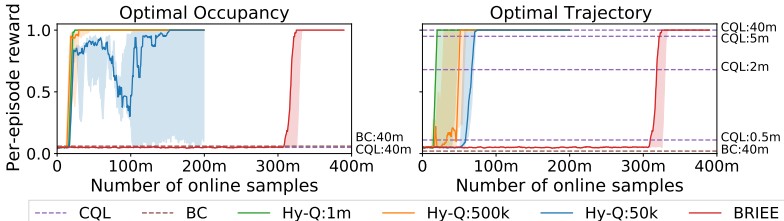

Figure 2: The learning curve for combination lock with $H = 100$. The plots show the median and 80th/20th quantile for 5 replicates. Pure offline and IL methods are visualized as dashed horizontal lines (in the left plot, CQL overlaps with BC). Note that we report the number of samples while Zhang et al. (2022c) report the number of episodes.

setup and also to compare with a provably efficient baseline (Zhang et al., 2022c). To study the third question, we consider the Montezuma's Revenge benchmark from the Arcade Learning environment, which is one of the most challenging empirical benchmarks with high-dimensional image inputs, largely due to the difficulties of exploration. Additional details are deferred to Appendix E.

**Hy-Q implementation.** We largely follow Algorithm 1 in our implementation for the combination lock experiment. Particularly, we use a similar function approximation to Zhang et al. (2022c), and a minibatch Adam update on Eq. (1) with the same sampling proportions as in the pseudocode. For Montezuma's Revenge, in addition to minibatch optimization, since the horizon of the environment is not fixed, we deploy a discounted version of Hy-Q. Concretely, the target value in the Bellman error is calculated from the output of a target network, which is periodically updated, times a discount factor. We refer the readers to Appendix E for more details.

**Baselines.** We include representative algorithms from four categories: (1) for imitation learning we use Behavior Cloning (BC) (Bain & Sammut, 1995), (2) for offline RL we use Conservative Q-Learning (CQL) (Kumar et al., 2020), (3) for online RL we use BRIEE (Zhang et al., 2022c) for combination lock[4] and Random Network Distillation (RND) (Burda et al., 2018) for Montezuma's Revenge, and (4) as a Hybrid-RL baseline we use Deep Q-learning from Demonstrations (DQFD) (Hester et al., 2018). We note that DQFD and prior hybrid RL methods combine expert demonstrations with online interactions, but are not necessarily designed to work with general offline datasets.

## 6.1 COMBINATION LOCK

The combination lock benchmark is depicted in Figure 3 and consists of horizon $H = 100$, three latent states for each time step and 10 actions in each state. Each state has a single "good" action that advances down a chain of favorable states from which optimal reward can be obtained. A single incorrect action transitions to an absorbing chain with suboptimal value. The agent operates on high dimensional observations and must use function approximation to succeed. This is an extremely challenging problem for which many Deep RL methods are known to fail (Misra et al., 2020), in part because (uniform) random exploration only has $10^{-H}$ probability of obtaining the optimal reward.

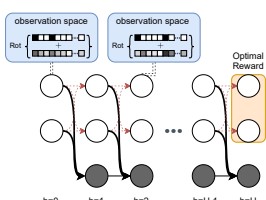

Figure 3: The combination lock (Zhang et al., 2022c), reproduced with permission.

On the other hand, the model has low bilinear rank, so we do have online RL algorithms that are provably sample-efficient: BRIEE currently obtains state of the art sample complexity. However, its sample complexity is still quite large, and we hope that Hybrid RL can address this shortcoming. We are not aware of any experiments with offline RL methods on this benchmark.

We construct two offline datasets for the experiments, both of which are derived from the optimal policy $\pi^\star$. In the **optimal trajectory** dataset we collect full trajectories by following $\pi^\star$ with $\epsilon$-greedy exploration with $\epsilon = 1/H$. In the **optimal occupancy** dataset we collect transition tuples from the state-occupancy measure of $\pi^\star$ with random actions.[5] Both datasets have bounded concentrability coefficients (and hence transfer coefficients) with respect to $\pi^\star$, but the second dataset is much more challenging since the actions do not directly provide information about $\pi^\star$, as they do in the former.

---

[4]We note that BRIEE is currently the state-of-the-art method for the combination lock environment. In particular, Misra et al. (2020) show that many Deep RL baselines fail in this enviroment.

[5]Formally, we sample $h \sim \text{Unif}([H])$, $s \sim d_h^{\pi^\star}$, $a \sim \text{Unif}(\mathcal{A})$, $r \sim R(s, a)$, $s' \sim P(s, a)$.

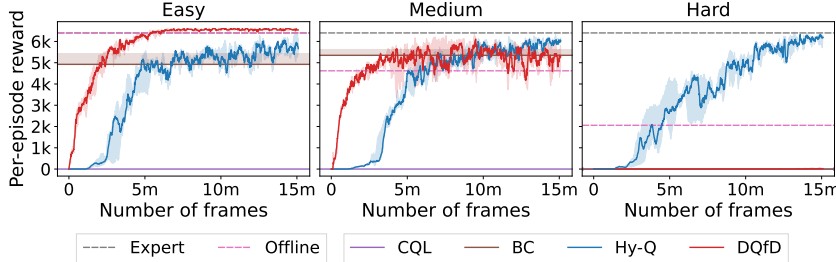

Figure 4: The learning curve for Montezuma's Revenge. The plots show the median and 80th/20th quantile for 5 replicates. Pure offline, IL methods and dataset qualities are visualized as dashed horizontal lines. "Expert" denotes $V^{\pi^e}$ and "Offline" denotes the average trajectory reward in the offline dataset. The y-axis denotes the (moving) average of 100 episodes for the methods involving online interactions. Note that CQL and BC overlap on the last plot.

The results are presented in Figure 2. First, we observe that Hy-Q can reliably solve the task under both offline distributions with relatively low sample complexity (500k offline samples and $\leq$ 25m online samples). In comparison, BC fails completely since both datasets contain random actions. CQL can solve the task using the trajectory-based dataset with a sample complexity that is comparable to the combined sample size of Hy-Q. However, CQL fails on the occupancy-based dataset since the actions themselves are not informative. Indeed the pessimism-inducing regularizer of CQL is constant on this dataset and so the algorithm reduces to FQI. Finally, Hy-Q can solve the task with a factor of 5-10 reduction in (online and offline) samples when compared with BRIEE. This demonstrates the robustness and sample efficiency provided by hybrid RL.

## 6.2 MONTEZUMA'S REVENGE

To answer the third question, we turn to Montezuma's Revenge, an extremely challenging image-based benchmark environment with sparse rewards. We follow the setup from Burda et al. (2018) and introduce stochasticity to the original dynamics: with probability 0.25 the environment executes the previous action instead of the current one. For offline datasets, we first train an "expert policy" $\pi^e$ via RND to achieve $V^{\pi^e} \approx 6400$. We create three datasets by mixing samples from $\pi^e$ with those from a random policy: the **easy dataset** contains only samples from $\pi^e$, the **medium dataset** mixes in a 80/20 proportion (80 from $\pi^e$), and the **hard dataset** mixes in a 50/50 proportion. Here we record full trajectories from both policies in the offline dataset, but measure the proportion using the number of transition tuples instead of trajectories. We provide 0.1 million offline samples for the hybrid methods, and 1 million samples for the offline and IL methods.

Results are displayed in Figure 4. CQL fails completely on all datasets. DQFD performs well on the easy dataset due to the large margin loss (Piot et al., 2014) that imitates the policies in the offline dataset. However, DQFD's performance drops as the quality of the offline dataset degrades (medium), and fails when the offline dataset is low quality (hard). We also observe that BC is a competitive baseline in the first two settings, and thus we view these problems as relatively easy to solve. Hy-Q is the only method that performs well on the hard dataset. Note that here, BC's performance is quite poor. We also include the comparison with RND in Figure 5: with only 100k offline samples from any of the three datasets, Hy-Q is over 10x more efficient in terms of online sample complexity.

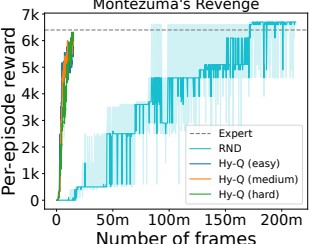

Figure 5: Learning curves of Hy-Q and RND. Metric follows Figure 4.

## 7 CONCLUSION

We demonstrate the potential of hybrid RL with Hy-Q, a simple, theoretically principled, and empirically effective algorithm. Our theoretical results showcase how Hy-Q circumvents the computational issues of pure offline or online RL, while our empirical results highlight its robustness and sample efficiency. Yet, Hy-Q is perhaps the most natural hybrid algorithm, and we are optimistic that there is much more potential to unlock from the hybrid setting. We look forward to studying this in the future.

**Reproducibility Statement.** For our theory results, we provide detailed proof in the Appendices. For experiments, we submit anonymous code in the supplemental materials. Our (offline) dataset can be reproduced with the attached instructions, and our results could be reproduced with the given random seeds. For more details, we include implementation, environment and computation hardware details in the Appendices, along with hyperparameters for both our method and the baselines. We also open source our code at https://github.com/yudasong/HyQ.

ACKNOWLEDGEMENT

AS thanks Karthik Sridharan for useful discussions. WS acknowledges funding support from NSF IIS-2154711. We thank Simon Zhao for their careful reading of the manuscript and improvement on the technical correctness of our paper. We also thank Uri Sherman for their discussion on the computational efficiency of the original draft.

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

## A   PROOFS FOR SECTION 5

**Additional notation.**   Throughout the appendix, we define the feature covariance matrix $\Sigma_{t;h}$ as

$$\Sigma_{t;h} = \sum_{\tau=1}^{t} X_h(f^\tau)(X_h(f^\tau))^\top + \lambda \mathbb{I}. \tag{3}$$

Furthermore, given a distribution $\beta \in \Delta(\mathcal{S} \times \mathcal{A})$ and a function $f : \mathcal{S} \times \mathcal{A} \mapsto \mathbb{R}$, we denote its weighted $\ell_2$ norm as $\|f\|_{2,\beta}^2 := \sqrt{\mathbb{E}_{s,a\sim\beta} f^2(s,a)}$.

### A.1   SUPPORTING LEMMAS FOR THEOREM 1

Before proving Theorem 1, we first present a few useful lemma. We start with a standard result on least square generalization bound, which is be used by recalling that Algorithm 1 performs least squares on the empirical bellman error. We defer the proof of Lemma 2 to Appendix B.

**Lemma 2.** *(Least squares generalization bound) Let $R > 0$, $\delta \in (0, 1)$, we consider a sequential function estimation setting, with an instance space $\mathcal{X}$ and target space $\mathcal{Y}$. Let $\mathcal{H} : \mathcal{X} \mapsto [-R, R]$ be a class of real valued functions. Let $\mathcal{D} = \{(x_1, y_1), \ldots, (x_T, y_T)\}$ be a dataset of $T$ points where $x_t \sim \rho_t := \rho_t(x_{1:t-1}, y_{1:t-1})$, and $y_t$ is sampled via the conditional probability $p(\cdot \mid x_t)$:*

$$y_t \sim p(\cdot \mid x_t) := h^*(x_t) + \varepsilon_t,$$

*where the function $h^*$ satisfies approximate realizability i.e.*

$$\inf_{h\in\mathcal{H}} \frac{1}{T} \sum_{t=1}^{T} \mathbb{E}_{x\sim\rho_t} \Big[ (h^*(x) - h(x))^2 \Big] \leq \gamma,$$

*and $\{\epsilon_i\}_{i=1}^{n}$ are independent random variables such that $\mathbb{E}[y_t \mid x_t] = h^*(x_t)$. Additionally, suppose that $\max_t |y_t| \leq R$ and $\max_x |h^*(x)| \leq R$. Then the least square solution $\widehat{h} \leftarrow \operatorname{argmin}_{h\in\mathcal{H}} \sum_{t=1}^{T} (h(x_t) - y_t)^2$ satisfies with probability at least $1 - \delta$,*

$$\sum_{t=1}^{T} \mathbb{E}_{x\sim\rho_t} \Big[ (\widehat{h}(x) - h^*(x))^2 \Big] \leq 3\gamma T + 256 R^2 \log(2|\mathcal{H}|/\delta).$$

The above lemma is basically an extension of the standard least square regression agnostic generalization bound from i.i.d. setting to the non-i.i.d. case with the sequence of training data forms a sequence of Martingales. We state the result when the realizability only holds approximately upto the approximation $\gamma$. However, for all our proofs, we invoke this result by setting $\gamma = 0$.

In the next two lemmas, we prove two lemmas where we can bound each part of the regret decomposition using the Bellman error of the value function $f$.

**Lemma 3** (Performance difference lemma). *For any function $f = (f_0, \ldots, f_{H-1})$ where $f_h : \mathcal{S} \times \mathcal{A} \mapsto \mathbb{R}$ and $h \in [H-1]$, we have*

$$\mathbb{E}_{s\sim d_0}[\max_a f_0(s,a) - V_0^{\pi^f}(s)] \leq \sum_{h=0}^{H-1} \Big| \mathbb{E}_{s,a\sim d_h^{\pi^f}}[f_h(s,a) - \mathcal{T}f_{h+1}(s,a)] \Big|,$$

*where we define $f_H(s,a) = 0$ for all $s, a$.*

*Proof.* We start the proof by noting that $\pi_0^f(s) = \operatorname{argmax}_a f_0(s,a)$, then we have:

$$\mathbb{E}_{s\sim d_0}[\max_a f_0(s,a) - V^{\pi^f}(s)] = \mathbb{E}_{s\sim d_0}[\mathbb{E}_{a\sim\pi_0^f(s)} f_0(s,a) - V_0^{\pi^f}(s)]$$

$$= \mathbb{E}_{s\sim d_0}[\mathbb{E}_{a\sim\pi_0^f(s)} f_0(s,a) - \mathcal{T}f_1(s,a)] + \mathbb{E}_{s\sim d_0}[\mathbb{E}_{a\sim\pi_0^f(s)} \mathcal{T}f_1(s,a) - V_0^{\pi^f}(s)]$$

$$= \mathbb{E}_{s,a\sim d_0^{\pi^f}}[f_0(s,a) - \mathcal{T}f_1(s,a)] +$$

$$\mathbb{E}_{s\sim d_0}[\mathbb{E}_{a\sim\pi_0^f(s)}[R(s,a) + \gamma\mathbb{E}_{s'\sim\mathcal{P}(s,a)} \max_{a'} f_1(s',a') - R(s,a) + \mathbb{E}_{s'\sim\mathcal{P}(s,a)} V_1^{\pi^f}(s')]]$$

$$= \mathbb{E}_{s,a \sim d_0^{\pi^f}}[f_0(s,a) - \mathcal{T}f_1(s,a)] + \mathbb{E}_{s \sim d_1^{\pi^f}}[\max_a f_1(s,a) - V_1^{\pi^f}(s)] \tag{4}$$

Then by recursively applying the same procedure on the second term in (4), we have

$$\mathbb{E}_{s \sim d_0}[\max_a f_0(s,a) - V^{\pi^f}(s)] = \sum_{h=0}^{H-1} \mathbb{E}_{s,a \sim d_h^{\pi^f}}[f_h(s,a) - \mathcal{T}f_{h+1}(s,a)] + \mathbb{E}_{s \sim d_H^{\pi^f}}[\max_a f_H(s,a) - V_H^{\pi^f}(s)].$$

Finally for $h = H$, we recall that we set $f_H(s,a) = 0$ and $V_H^{\pi^f} = 0$ for notation simplicity. Thus we have:

$$\mathbb{E}_{s \sim d_0}[\max_a f_0(s,a) - V^{\pi^f}(s)] = \sum_{h=0}^{H-1} \mathbb{E}_{s,a \sim d_h^{\pi^f}}[f_h(s,a) - \mathcal{T}f_{h+1}(s,a)]$$

$$\leq \sum_{h=0}^{H-1} \left| \mathbb{E}_{s,a \sim d_h^{\pi^f}}[f_h(s,a) - \mathcal{T}f_{h+1}(s,a)] \right|.$$

$\square$

Now we proceed to how to bound the other half in the regret decomposition:

**Lemma 4.** *Let $\pi^e = (\pi_0^e, \ldots, \pi_{H-1}^e)$ be a comparator policy, and consider any value function $f = (f_0, \ldots, f_{H-1})$ where $f_h : \mathcal{S} \times \mathcal{A} \mapsto \mathbb{R}$. Then,*

$$\mathbb{E}_{s \sim d_0}\left[V_0^{\pi^e}(s) - \max_a f_0(s,a)\right] \leq \sum_{i=0}^{H-1} \mathbb{E}_{s,a \sim d_i^{\pi^e}}[\mathcal{T}f_{i+1}(s,a) - f_i(s,a)],$$

*where we defined $f_H(s,a) = 0$ for all $s,a$.*

*Proof.* The proof is similar to the proof of Lemma 3, and we start with the fact that $\max_a f(s,a) \geq f(s,a'), \forall a'$, including actions sampled from $\pi^e$:

$$\mathbb{E}_{s \sim d_0}\left[V_0^{\pi^e}(s) - \max_a f_0(s,a)\right] \leq \mathbb{E}_{s,a \sim d_0^{\pi^e}}\left[Q_0^{\pi^e}(s,a) - f_0(s,a)\right]$$

$$= \mathbb{E}_{s,a \sim d_0^{\pi^e}}\left[Q_0^{\pi^e}(s,a) - \mathcal{T}f_1(s,a) + \mathcal{T}f_1(s,a) - f_0(s,a)\right]$$

$$= \mathbb{E}_{s,a \sim d_0^{\pi^e}}\left[\mathbb{E}_{s' \sim \mathcal{P}(s,a)}V_1^{\pi^e}(s') - \max_{a'} f_1(s',a')\right] + \mathbb{E}_{s,a \sim d_0^{\pi^e}}[\mathcal{T}f_1(s,a) - f_0(s,a)]$$

$$= \mathbb{E}_{s \sim d_1^{\pi^e}}\left[V_1^{\pi^e}(s) - \max_a f_1(s,a)\right] + \mathbb{E}_{s,a \sim d_0^{\pi^e}}[\mathcal{T}f_1(s,a) - f_0(s,a)] \tag{5}$$

Again by recursively applying the same procedure on the first term in (5), we have

$$\mathbb{E}_{s \sim d_0}\left[V_0^{\pi^e}(s) - \max_a f_0(s,a)\right] \leq \mathbb{E}_{s \sim d_H^{\pi^e}}\left[V_H^{\pi^e}(s) - \max_a f_H(s,a)\right] + \sum_{h=0}^{H-1} \mathbb{E}_{s,a \sim d_h^{\pi^e}}[\mathcal{T}f_{h+1}(s,a) - f_h(s,a)],$$

and recall that $f_H(s,a) = 0$ and $V_H^{\pi^f} = 0$, we have

$$\mathbb{E}_{s \sim d_0}\left[V_0^{\pi^e}(s) - \max_a f_0(s,a)\right] \leq \sum_{h=0}^{H-1} \mathbb{E}_{s,a \sim d_h^{\pi^e}}[\mathcal{T}f_{h+1}(s,a) - f_h(s,a)].$$

$\square$

The following result is useful in the bilinear models when we want to bound the potential functions. The result directly follows from the elliptical potential lemma (Lattimore & Szepesvári, 2020, Lemma 19.4).

**Lemma 5.** *Let $X_h(f^1), \ldots, X_h(f^T) \in \mathbb{R}^d$ be a sequence of vectors with $\|X_h(f^t)\| \leq B_X < \infty$ for all $t \leq T$. Then,*

$$\sum_{t=1}^{T} \|X_h(f^t)\|_{\Sigma_{t-1;h}^{-1}} \leq \sqrt{2dT \log\left(1 + \frac{TB_X^2}{\lambda d}\right)},$$

*where the matrix $\Sigma_{t;h} := \sum_{\tau=1}^{t} X_h(f^\tau)X_h(f^\tau)^\top + \lambda\mathbb{I}$ for $t \in [T]$ and $\lambda \geq B_X^2$, and the matrix norm $\|X_h(f^t)\|_{\Sigma_{t-1;h}^{-1}} = \mathbb{E}[X_h(f^t)^\top \Sigma_{t-1;h}^{-1} X_h(f^t)]$.*

*Proof.* Since $\lambda \geq B_X^2$, we have that

$$\|X_h(f^t)\|_{\Sigma_{t-1;h}^{-1}}^2 \leq \frac{1}{\lambda}\|X_h(f^t)\|^2 \leq 1.$$

Thus, using elliptical potential lemma (Lattimore & Szepesvári, 2020, Lemma 19.4), we get that

$$\sum_{t=1}^{T} \|X_h(f^t)\|_{\Sigma_{t-1;h}^{-1}}^2 \leq 2d \log\left(1 + \frac{TB_X^2}{\lambda d}\right).$$

The desired bound follows from Jensen's inequality which implies that

$$\sum_{t=1}^{T} \|X_h(f^t)\|_{\Sigma_{t-1;h}^{-1}} \leq \sqrt{T \cdot \sum_{t=1}^{T} \|X_h(f^t)\|_{\Sigma_{t-1;h}^{-1}}^2} \leq \sqrt{2Td \log\left(1 + \frac{TB_X^2}{\lambda d}\right)}.$$

$\square$

## A.2 PROOF OF THEOREM 1

Before delving into the proof, we first state that following generalization bound for FQI.

**Lemma 6** (Bellman error bound for FQI)**.** *Let $\delta \in (0, 1)$ and let for $h \in [H-1]$ and $t \in [T]$, $f_h^{t+1}$ be the estimated value function for time step $h$ computed via least square regression using samples in the dataset $\left(\mathcal{D}_h^\nu, \mathcal{D}_h^1, \ldots, \mathcal{D}_h^t\right)$ in (1) in the iteration $t$ of Algorithm 1. Then, with probability at least $1 - \delta$, for any $h \in [H-1]$ and $t \in [T]$,*

$$\left\|f_h^{t+1} - \mathcal{T}f_{h+1}^{t+1}\right\|_{2,\nu_h}^2 \leq \frac{1}{m_{\text{off}}} 256V_{\max}^2 \log(2HT|\mathcal{F}|/\delta) =: \Delta_{\text{off}},$$

*and*

$$\sum_{\tau=1}^{t} \left\|f_h^{t+1} - \mathcal{T}f_{h+1}^{t+1}\right\|_{2,\mu_h^\tau}^2 \leq \frac{1}{m_{\text{on}}} 256V_{\max}^2 \log(2HT|\mathcal{F}|/\delta) =: \Delta_{\text{on}},$$

*where $\nu_h$ denotes the offline data distribution at time $h$, and the distribution $\mu_h^\tau \in \Delta(s, a)$ is defined such that $s, a \sim d_h^{\pi^\tau}$.*

*Proof.* Fix $t \in [T]$, $h \in [H-1]$ and $f_{h+1}^{t+1} \in \mathcal{F}_{h+1}$ and consider the regression problem ((1) in the iteration $t$ of Algorithm 1):

$$f_h^{t+1} \leftarrow \underset{f \in \mathcal{F}_h}{\operatorname{argmin}}\left\{\widehat{\mathbb{E}}_{\mathcal{D}_h^t}(f(s, a) - r - \max_{a'} f_{h+1}^{t+1}(s', a'))^2\right\},$$

where dataset $\mathcal{D}_h^t = \mathcal{D}_h^\nu + \sum_{\tau=1}^{t} \mathcal{D}_h^\tau$ consisting of $n = m_{\text{off}} + t \cdot m_{\text{on}}$ samples $\{(x_i, y_i)\}_{i \leq n}$ where

$$x_i = (s_h^i, a_h^i) \qquad \text{and} \qquad y^i = r^i + \max_a f_{h+1}^{t+1}(s_{h+1}^i, a).$$

In particular, we define $\mathcal{D}$ such that the first $m_{\text{off}}$ samples $\{(x_i, y_i)\}_{i \leq m_{\text{off}}} = \mathcal{D}_h^\nu$, the next $m_{\text{on}}$ samples $\{(x_i, y_i)\}_{i=m_{\text{off}}+1}^{m_{\text{off}}+m_{\text{on}}} = \mathcal{D}_h^1$, and so on where the samples $\{(x_i, y_i)\}_{i=m_{\text{off}}+(\tau-1)m_{\text{on}}+1}^{m_{\text{off}}+\tau m_{\text{on}}} = \mathcal{D}_h^\tau$. Note that: (a) for any sample $(x = (s_h, a_h), y = (r + \max_a f_{h+1}^{t+1}(s_{h+1}, a)))$ in $\mathcal{D}$, we have that

$$\mathbb{E}[y \mid x] = \mathbb{E}_{s_{h+1} \sim P(s_h, a_h), r \sim R(s_h, a_h)}\left[r + \max_a f_{h+1}^{t+1}(s_{h+1}, a)\right]$$

$$= \mathcal{T} f_{h+1}^{t+1}(s_h, a_h) \le g(s_h, a_h),$$

where the last line holds since the Bellman completeness assumption implies existence of such a function $g$, (b) for any sample, $|y| \le V_{\max}$ and $f(s, a) \le V_{\max}$ for all $s, a$, (c) our construction of $\mathcal{D}$ implies that for each iteration $t$, the sample $(x_t, y_t)$ are generated in the following procedure: $x_t$ is sampled from the data generation scheme $\mathcal{D}^t(x_{1:t-1}, y_{1:t-1})$, and $y_t$ is sampled from some conditional probability distribution $p(\cdot \mid x_t)$ as defined in Lemma 2, finally (d) the samples in $\mathcal{D}_h^\nu$ are drawn from the offline distribution $\nu_h$, and the samples in $\mathcal{D}_h^\tau$ are drawn such that $s_h \sim d_h^{\pi^t}$ and $a_h \sim \pi^{f^t}(s_h)$. Thus, using Lemma 2, we get that the least square regression solution $f_h^{t+1}$ satisfies

$$\sum_{i=1}^{n} \mathbb{E}\big[(f_h^{t+1}(s^i, a^i) - \mathcal{T} f_{h+1}^{t+1}(s^i, a^i))^2 \mid \mathcal{D}_i\big] \le 256 V_{\max}^2 \log(2|\mathcal{F}|/\delta).$$

Using the property-(d) in the above, we get that

$$m_{\mathrm{off}} \cdot \big\| f_h^{t+1} - \mathcal{T} f_{h+1}^{t+1} \big\|_{2, \nu_h}^2 + m_{\mathrm{on}} \cdot \sum_{\tau=1}^{t} \big\| f_h^{t+1} - \mathcal{T} f_{h+1}^{t+1} \big\|_{2, \mu_h^\tau}^2 \le 256 V_{\max}^2 \log(2|\mathcal{F}|/\delta),$$

where the distribution $\mu_h^\tau \in \Delta(s, a)$ is defined by sampling $s \sim d_h^{\pi^\tau}$ and $a \sim \pi^{f^t}(s)$. Taking a union bound over $h \in [H-1]$ and $t \in [T]$, and bounding each term separately, gives the desired statement. $\qquad\square$

We next note a change in distribution lemma which allows us to bound expected bellman error under the $(s, a)$ distribution generated by $f^t$ in terms of the expected square bellman error w.r.t. the previous policies data distribution, which is further controlled using regression.

**Lemma 7.** *For any $t \ge 0$ and $h \in [H-1]$, we have*

$$\big|\langle W_h(f^t), X_h(f^t)\rangle\big| \le \|X_h(f^t)\|_{\Sigma_{t-1;h}^{-1}} \sqrt{\sum_{i=1}^{t-1} \mathbb{E}_{s,a \sim d_h^{f^i}}\Big[\big(f_h^t - \mathcal{T} f_{h+1}^t\big)^2\Big] + \lambda B_W^2},$$

*where $\Sigma_{t-1}^{-1}$ is defined in (3) and use the notation $d_h^{f^i}$ to denote $d_h^{\pi^{f^i}}$.*

*Proof.* Using Cauchy-Schwarz inequality, we get that

$$\big|\langle W_h(f^t), X_h(f^t)\rangle\big| \le \|X_h(f^t)\|_{\Sigma_{t-1;h}^{-1}} \|W_h(f^t)\|_{\Sigma_{t-1;h}}$$

$$= \|X_h(f^t)\|_{\Sigma_{t-1;h}^{-1}} \sqrt{(W_h(f^t))^\top \Sigma_{t-1} W_h(f^t)}$$

$$= \|X_h(f^t)\|_{\Sigma_{t-1;h}^{-1}} \sqrt{(W_h(f^t))^\top \left(\sum_{i=1}^{t-1} X_h(f^i) X_h(f^i)^\top + \lambda \mathbb{I}\right) W_h(f^t)}$$

$$= \|X_h(f^t)\|_{\Sigma_{t-1;h}^{-1}} \sqrt{\sum_{i=1}^{t-1} |\langle W_h(f^t), X_h(f^i)\rangle|^2 + \lambda \|W_h(f^t)\|^2}$$

$$\le \|X_h(f^t)\|_{\Sigma_{t-1;h}^{-1}} \sqrt{\sum_{i=1}^{t-1} |\langle W_h(f^t), X_h(f^i)\rangle|^2 + \lambda B_W^2} \qquad (6)$$

$$\le \|X_h(f^t)\|_{\Sigma_{t-1;h}^{-1}} \sqrt{\sum_{i=1}^{t-1} \mathbb{E}_{s,a \sim d_h^{f^i}}\Big[\big(f_h^t - \mathcal{T} f_{h+1}^t\big)^2\Big] + \lambda B_W^2}$$

where the inequality in the second last line holds by plugging in the bound on $\|W_h(f^t)\|$, and the last line holds by using Definition 2 which implies that

$$\big|\langle W_h(f^t), X_h(f^i)\rangle\big|^2 = \Big(\mathbb{E}_{s,a \sim d_h^{f^i}}\big[f_h^t - \mathcal{T} f_{h+1}^t\big]\Big)^2 \le \mathbb{E}_{s,a \sim d_h^{f^i}}\Big[\big(f_h^t - \mathcal{T} f_{h+1}^t\big)^2\Big],$$

where the last inequality is due to Jensen's inequality. $\qquad\square$

We now have all the tools to prove Theorem 1. We first restate the bound with the exact problem dependent parameters, assumign that $B_W$ and $B_X$ are constants which are hidden in the order notation below.

**Theorem** (Theorem 1 restated). *Let* $m_{\text{off}} = T$ *and* $m_{\text{on}} = 1$. *Then, with probability at least* $1 - \delta$, *the cumulative suboptimality of Algorithm 1 is bounded as*

$$\sum_{t=1}^{T} V^{\pi^e} - V^{\pi^{f^t}} = O\left( \max\{C_{\pi^e}, 1\} V_{\max} \sqrt{dH^2 T \cdot \log\left(1 + \frac{T}{d}\right) \log\left(\frac{HT|\mathcal{F}|}{\delta}\right)} \right).$$

*Proof of Theorem 1.* Let $\pi^e$ be any comparator policy with bounded transfer coefficient i.e.

$$C_{\pi^e} := \max \left\{ 0, \ \max_{f \in \mathcal{F}} \frac{\sum_{h=0}^{H-1} \mathbb{E}_{s,a \sim d_h^{\pi^e}} [f_h(s,a) - \mathcal{T}f_{h+1}(s,a)]}{\sqrt{\sum_{h=0}^{H-1} \mathbb{E}_{s,a \sim \nu_h} \left[ (f_h(s,a) - \mathcal{T}f_{h+1}(s,a))^2 \right]}} \right\} < \infty. \tag{7}$$

We start by noting that

$$\sum_{t=1}^{T} V^{\pi^e} - V^{\pi^{f^t}} = \sum_{t=1}^{T} \mathbb{E}_{s \sim d_0} \left[ V_0^{\pi^e}(s) - V_0^{\pi^{f^t}}(s) \right]$$

$$= \sum_{t=1}^{T} \mathbb{E}_{s \sim d_0} \left[ V_0^{\pi^e}(s) - \max_a f_0^t(s,a) \right] + \sum_{t=1}^{T} \mathbb{E}_{s \sim d_0} \left[ \max_a f_0^t(s,a) - V_0^{\pi^{f^t}}(s) \right]. \tag{8}$$

For the first term in the right hand side of (8), note that using Lemma 4 for each $f_t$ for $1 \le t \le T$, we get

$$\sum_{t=1}^{T} \mathbb{E}_{s \sim d_0} \left[ V_0^{\pi^e}(s) - \max_a f_0^t(s,a) \right] \le \sum_{t=1}^{T} \sum_{h=0}^{H-1} \mathbb{E}_{s,a \sim d_h^{\pi^e}} [\mathcal{T}f_{h+1}^t(s,a) - f_h^t(s,a)]$$

$$\le \sum_{t=1}^{T} C_{\pi^e} \cdot \sqrt{\sum_{h=0}^{H-1} \mathbb{E}_{s,a \sim \nu_h} \left[ (f_h^t(s,a) - \mathcal{T}f_{h+1}^t(s,a))^2 \right]}$$

$$= TC_{\pi^e} \cdot \sqrt{H \cdot \Delta_{\text{off}}}, \tag{9}$$

where the second inequality follows from plugging in the definition of $C_{\pi_e}$ in (7). The last line follows from Lemma 6.

For the second term in (8), using Lemma 3 for each $f_t$ for $1 \le t \le T$, we get

$$\sum_{t=1}^{T} \mathbb{E}_{s \sim d_0} \left[ \max_a f_0^t(s,a) - V_0^{\pi^{f^t}}(s) \right] \le \sum_{t=1}^{T} \sum_{h=0}^{H-1} \left| \mathbb{E}_{s,a \sim d_h^{\pi^{f^t}}} \left[ f_h^t(s,a) - \mathcal{T}f_{h+1}^t(s,a) \right] \right| \tag{10}$$

$$= \sum_{t=1}^{T} \sum_{h=0}^{H-1} \left| \langle X_h(f^t), W_h(f^t) \rangle \right|$$

$$\le \sum_{t=1}^{T} \sum_{h=0}^{H-1} \|X_h(f^t)\|_{\Sigma_{t-1;h}^{-1}} \sqrt{\Delta_{\text{on}} + \lambda B_W^2},$$

where the second line follows from Definition 2, the third line follows from Lemma 7 and by plugging in the bound in Lemma 6. Using the bound in Lemma 5 in the above, we get that

$$\sum_{t=1}^{T} \mathbb{E}_{s \sim d_0} \left[ \max_a f_0^t(s,a) - V_0^{\pi^{f^t}}(s) \right] \le \sqrt{2dH^2 \log\left(1 + \frac{TB_X^2}{\lambda d}\right) \cdot (\Delta_{\text{on}} + \lambda B_W^2) \cdot T}$$

$$\le \sqrt{2dH^2 \log\left(1 + \frac{T}{d}\right) \cdot (\Delta_{\text{on}} + B_X^2 B_W^2) \cdot T}, \tag{11}$$

where the second line follows by plugging in $\lambda = B_X^2$.

Combining the bound (9) and (11), we get that

$$\sum_{t=1}^{T} V^{\pi^e} - V^{\pi^{f^t}} \le TC_{\pi^e} \cdot \sqrt{H \cdot \Delta_{\text{off}}} + \sqrt{2dH^2 \log\left(1 + \frac{T}{d}\right) \cdot (\Delta_{\text{on}} + B_X^2 B_W^2) \cdot T}$$

Plugging in the values of $\Delta_{\text{on}}$ and $\Delta_{\text{off}}$ in the above, and using subadditivity of square-root, we get that

$$\sum_{t=1}^{T} V^{\pi^e} - V^{\pi^{f^t}} \le 16 V_{\max} C_{\pi^e} T \sqrt{\frac{H}{m_{\text{off}}} \log\left(\frac{2HT|\mathcal{F}|}{\delta}\right)} + 16 V_{\max} \sqrt{\frac{2dH^2T}{m_{\text{on}}} \log\left(1 + \frac{T}{d}\right) \log\left(\frac{2HT|\mathcal{F}|}{\delta}\right)}$$

$$+ HB_X B_W \sqrt{2dT \log\left(1 + \frac{T}{d}\right)}.$$

Setting $m_{\text{off}} = T$ and $m_{\text{on}} = 1$ in the above gives the cumulative suboptimality bound

$$\sum_{t=1}^{T} V^{\pi^e} - V^{\pi^{f^t}} = O\left(\max\{C_{\pi^e}, 1\} V_{\max} \sqrt{dH^2 T \cdot \log\left(1 + \frac{T}{d}\right) \log\left(\frac{HT|\mathcal{F}|}{\delta}\right)}\right). \qquad (12)$$

$\square$

*Proof of Corollary 1.* We next convert the above cumulative suboptimality bound into sample complexity bound via a standard online-to-batch conversion. Setting $\pi^e = \pi^*$ in (12) and defining the policy $\hat{\pi} = \text{Uniform}(\{\pi^1, \ldots, \pi^T\})$, we get that

$$\mathbb{E}\left[V^{\pi^*} - V^{\hat{\pi}}\right] = \frac{1}{T}\left(\sum_{t=1}^{T} V^{\pi^*} - V^{\pi^t}\right)$$

$$= O\left(\max\{C_{\pi^*}, 1\} V_{\max} \sqrt{\frac{dH^2}{T} \cdot \log\left(1 + \frac{T}{d}\right) \log\left(\frac{HT|\mathcal{F}|}{\delta}\right)}\right).$$

Thus, we get that for $T \ge \widetilde{O}\left(\frac{\max\{C_{\pi^*}^2, 1\} V_{\max}^2 dH^2 \log(HT|\mathcal{F}|/\delta)}{\epsilon^2}\right)$, we get that

$$\mathbb{E}\left[V^{\pi^*} - V^{\hat{\pi}}\right] \le \epsilon.$$

In these $T$ iterations, the total number of offline samples used is

$$m_{\text{off}} = T = \widetilde{O}\left(\frac{\max\{C_{\pi^*}^2, 1\} V_{\max}^2 dH^2 \log(HT|\mathcal{F}|/\delta)}{\epsilon^2}\right),$$

and the total number of online samples used is

$$m_{\text{on}} \cdot H \cdot T = \widetilde{O}\left(\frac{\max\{C_{\pi^*}^2, 1\} V_{\max}^2 dH^3 \log(HT|\mathcal{F}|/\delta)}{\epsilon^2}\right),$$

where the additional $H$ factor appears because we collect $m_{\text{on}}$ samples for every $h \in [H]$ in the algorithm. $\square$

## A.3 V-TYPE BILINEAR RANK

Our previous result focus on the Q-type bilinear model. Here we provide the V-type Bilinear rank definition. This V-type Bilinear rank definition is basically the same as the low Bellman rank model proposed by Jiang et al. (2017).

---

**Algorithm 2** V-type Hy-Q

---

**Require:** Value function class: $\mathcal{F}$, #iterations: $T$, Offline dataset $\mathcal{D}_h^\nu$ of size $m_{\text{off}}$ for $h \in [H-1]$.
1: Initialize $f_h^1(s,a) = 0$.
2: **for** $t = 1, \ldots, T$ **do**
3:     Let $\pi^t$ be the greedy policy w.r.t. $f^t$ i.e., $\pi_h^t(s) = \text{argmax}_a f_h^t(s,a)$.
      **// Online collection**
4:     For each $h$, collect $m_{\text{on}}$ online tuples $\mathcal{D}_h^t \sim d_h^{\pi^t} \circ \text{Uniform}(\mathcal{A})$.
      **// FQI using both online and offline data**
5:     Set $f_H^{t+1}(s,a) = 0$.
6:     **for** $h = H-1, \ldots, 0$ **do**
7:         Estimate $f_h^{t+1}$ using least squares regression on the aggregated data:

$$f_h^{t+1} \leftarrow \underset{f \in \mathcal{F}_h}{\text{argmin}} \left\{ \widehat{\mathbb{E}}_{\mathcal{D}_h^\nu}(f(s,a) - r - \max_{a'} f_{h+1}^{t+1}(s',a'))^2 + \sum_{\tau=1}^t \widehat{\mathbb{E}}_{\mathcal{D}_h^\tau}(f(s,a) - r - \max_{a'} f_{h+1}^{t+1}(s',a'))^2 \right\}$$
(13)

8:     **end for**
9: **end for**

---

**Definition 4** (V-type Bilinear model). *Consider any pair of functions $(f,g)$ with $f,g \in \mathcal{F}$. Denote the greedy policy of $f$ as $\pi^f = \{\pi_h^f := \text{argmax}_a f_h(s,a), \forall h\}$. We say that the MDP together with the function $\mathcal{F}$ admits a bilinear structure of rank $d$ if for any $h \in [H-1]$, there exist two (unknown) mappings $X_h : \mathcal{F} \mapsto \mathbb{R}^d$ and $W_h : \mathcal{F} \mapsto \mathbb{R}^d$ with $\max_f \|X_h(f)\|_2 \leq B_X$ and $\max_f \|W_h(f)\|_2 \leq B_W$, such that:*

$$\forall f, g \in \mathcal{F} : \left| \mathbb{E}_{s \sim d_h^{\pi^f}, a \sim \pi_g(s)} g_h(s,a) - \mathcal{T}g_{h+1}(s,a) \right| = |\langle X_h(f), W_h(g) \rangle|.$$

Note that different from the Q-type definition, here the action $a$ is taken from the greedy policy with respect to $g$. This way $\max_a g(s,a)$ can serve as an approximation of $V^\star$ – thus the name of $V$-type.

To make Hy-Q work for the V-type Bilinear model, we only need to make slight change on the data collection process, i.e., when we collect online batch $\mathcal{D}_h$, we sample $s \sim d_h^{\pi^t}$, $a \sim \text{Uniform}(\mathcal{A})$, $s' \sim P(\cdot|s,a)$. Namely the action is taken uniformly randomly here. We provide the pseudocode in Algorithm 2. We refer the reader to Du et al. (2021); Jin et al. (2021a) for a detailed discussion.

### A.3.1 COMPLEXITY BOUND FOR V-TYPE BILINEAR MODELS

In this section, we give a performance analysis of Algorithm 2 for V-type Bilinear models. The contents in this section extend the results developed for Q-type Bilinear models in Section A.2 to V-type Bilinear models.

We first note the following bound for FQI estimates in Algorithm 2.

**Lemma 8.** *Let $\delta \in (0,1)$ and let for $h \in [H-1]$ and $t \in [T]$, $f_h^{t+1}$ be the estimated value function for time step $h$ computed via least square regression using samples in the dataset $\left(\mathcal{D}_h^\nu, \mathcal{D}_h^1, \ldots, \mathcal{D}_h^t\right)$ in (13) in the iteration $t$ of Algorithm 2. Then, with probability at least $1 - \delta$, for any $h \in [H-1]$ and $t \in [T]$,*

$$\left\| f_h^{t+1} - \mathcal{T}f_{h+1}^{t+1} \right\|_{2,\nu_h}^2 \leq \frac{1}{m_{\text{off}}} 256 V_{\max}^2 \log(2HT|\mathcal{F}|/\delta) =: \bar{\Delta}_{\text{off}},$$

*and*

$$\sum_{\tau=1}^t \left\| f_h^{t+1} - \mathcal{T}f_{h+1}^{t+1} \right\|_{2,\mu_h^\tau}^2 \leq \frac{1}{m_{\text{on}}} 256 V_{\max}^2 \log(2HT|\mathcal{F}|/\delta) =: \bar{\Delta}_{\text{on}},$$

*where $\nu_h$ denotes the offline data distribution at time $h$, and the distribution $\mu_h^\tau \in \Delta(s,a)$ is defined such that $s \sim d_h^{\pi^\tau}$ and $a \sim \text{Uniform}(\mathcal{A})$.*

The following change in distribution lemma is the version of Lemma 7 under V-type Bellman rank assumption.

**Lemma 9.** *Suppose the underlying model is a V-type bilinear model. Then, for any $t \geq 0$ and $h \in [H-1]$, we have*

$$\left|\langle W_h(f^t), X_h(f^t)\rangle\right| \leq \|X_h(f^t)\|_{\Sigma_{t-1;h}^{-1}} \sqrt{|\mathcal{A}| \cdot \sum_{i=1}^{t-1} \mathbb{E}_{s \sim d_h^{\pi f^i}, \, a \sim \text{Uniform}(\mathcal{A})}\left[\left(f_h^t - \mathcal{T}f_{h+1}^t\right)^2\right] + \lambda B_W^2},$$

*where $\Sigma_{t-1}^{-1}$ is defined in* (3).

*Proof.* The proof is identical to the proof of Lemma 7. Repeating the analysis till (6), we get that

$$\left|\langle W_h(f^t), X_h(f^t)\rangle\right| \leq \|X_h(f^t)\|_{\Sigma_{t-1;h}^{-1}} \sqrt{\sum_{i=1}^{t-1} \left|\langle W_h(f^t), X_h(f^i)\rangle\right|^2 + \lambda B_W^2}$$

$$= \|X_h(f^t)\|_{\Sigma_{t-1;h}^{-1}} \sqrt{\sum_{i=1}^{t-1} \left(\mathbb{E}_{s \sim d_h^{\pi f^i}, \, a \sim \pi^{f^t}(s)}\left[f_h^t - \mathcal{T}f_{h+1}^t\right]\right)^2 + \lambda B_W^2}$$

$$\leq \|X_h(f^t)\|_{\Sigma_{t-1;h}^{-1}} \sqrt{|\mathcal{A}| \cdot \sum_{i=1}^{t-1} \mathbb{E}_{s \sim d_h^{\pi f^i}, \, a \sim \text{Uniform}(\mathcal{A})}\left[\left(f_h^t - \mathcal{T}f_{h+1}^t\right)^2\right] + \lambda B_W^2}$$

where the second line above follows from the definition of V-type bilinear model in Definition 4, and the last line holds because:

$$\left(\mathbb{E}_{s \sim d_h^{\pi f^i}, \, a \sim \pi^{f^t}(s)}\left[f_h^t - \mathcal{T}f_{h+1}^t\right]\right)^2 \leq \mathbb{E}_{s \sim d_h^{\pi f^i}, a \sim \pi^{f^t}(s)}\left[\left(f_h^t - \mathcal{T}f_{h+1}^t\right)^2\right]$$

$$\leq |\mathcal{A}| \cdot \mathbb{E}_{s \sim d_h^{\pi f^i}, a \sim \text{Uniform}(\mathcal{A})}\left[\left(f_h^t - \mathcal{T}f_{h+1}^t\right)^2\right]$$

where the first inequality above is due to Jensen's inequality and the last inequality follows form a straightforward upper bound since each term inside the expectation is non-negative. □

We are finally ready to state and prove our main result in this section.

**Theorem 2** (Cumulative suboptimality bound for V-type bilinear rank models)**.** *Let $m_{\text{on}} = |\mathcal{A}|$ and $m_{\text{off}} = T$. Then, with probability at least $1 - \delta$, the cumulative suboptimality of Algorithm 2 is bounded as*

$$\sum_{t=1}^{T} V^{\pi^e} - V^{\pi^{f^t}} = O\left(\max\{C_{\pi^e}, 1\}V_{\max}\sqrt{dH^2 T \cdot \log\left(1 + \frac{T}{d}\right)\log\left(\frac{HT|\mathcal{F}|}{\delta}\right)}\right)$$

*Proof.* The proof follows closely the proof of Theorem 1. Repeating the analysis till (8) and (9), we get that:

$$\sum_{t=1}^{T} V^{\pi^e} - V^{\pi^{f^t}} \leq TC_{\pi^e} \cdot \sqrt{H \cdot \bar{\Delta}_{\text{off}}} + \sum_{t=1}^{T} \mathbb{E}_{s \sim d_0}\left[\max_a f_0^t(s, a) - V_0^{\pi^{f^t}}(s)\right]. \quad (14)$$

For the second term in the above, using Lemma 3 for each $f_t$ for $1 \leq t \leq T$, we get

$$\sum_{t=1}^{T} \mathbb{E}_{s \sim d_0}\left[\max_a f_0^t(s, a) - V_0^{\pi^{f^t}}(s)\right] \leq \sum_{t=1}^{T} \sum_{h=0}^{H-1} \left|\mathbb{E}_{s, a \sim d_h^{\pi f^t}}\left[f_h^t(s, a) - \mathcal{T}_h f_{h+1}^t(s, a)\right]\right|$$

$$= \sum_{t=1}^{T} \sum_{h=0}^{H-1} \left|\langle X_h(f^t), W_h(f^t)\rangle\right|$$

$$\leq \sum_{t=1}^{T} \sum_{h=0}^{H-1} \|X_h(f^t)\|_{\Sigma_{t-1;h}^{-1}} \sqrt{|\mathcal{A}| \cdot \bar{\Delta}_{\mathrm{on}} + \lambda B_W^2},$$

where the second line follows from Definition 4, and the last line follows from Lemma 9 and by plugging in the bound in Lemma 8. Using the elliptical potential Lemma 5 as in the proof of Theorem 1, we get that

$$\sum_{t=1}^{T} V^{\pi^e} - V^{\pi^{f^t}} \leq TC_{\pi^e} \cdot \sqrt{H \cdot \bar{\Delta}_{\mathrm{off}}} + \sqrt{2dH^2 \log\left(1 + \frac{T}{d}\right) \cdot \left(|\mathcal{A}| \cdot \bar{\Delta}_{\mathrm{on}} + B_X^2 B_W^2\right) \cdot T}$$

Plugging in the values of $\bar{\Delta}_{\mathrm{on}}$ and $\bar{\Delta}_{\mathrm{off}}$ from Lemma 8 in the above, and using subadditivity of square-root, we get that

$$\sum_{t=1}^{T} V^{\pi^e} - V^{\pi^{f^t}} \leq 16 V_{\max} C_{\pi^e} T \sqrt{\frac{H}{m_{\mathrm{off}}} \log\left(\frac{2HT|\mathcal{F}|}{\delta}\right)} + 16 V_{\max} \sqrt{\frac{2dH^2|\mathcal{A}|T}{m_{\mathrm{on}}} \log\left(1 + \frac{T}{d}\right) \log\left(\frac{2HT|\mathcal{F}|}{\delta}\right)}$$
$$+ HB_X B_W \sqrt{2dT \log\left(1 + \frac{T}{d}\right)}.$$

Setting $m_{\mathrm{on}} = |\mathcal{A}|$ and $m_{\mathrm{off}} = T$, we get the following cumulative suboptimality bound:

$$\sum_{t=1}^{T} V^{\pi^e} - V^{\pi^{f^t}} = O\left(\max\{C_{\pi^e}, 1\} V_{\max} \sqrt{dH^2 T \cdot \log\left(1 + \frac{T}{d}\right) \log\left(\frac{HT|\mathcal{F}|}{\delta}\right)}\right). \quad (15)$$

$\square$

**Corollary 2** (Sample complexity). *Under the assumptions of Theorem 2 if $C_{\pi^*} < \infty$ then Algorithm 2 can find an $\epsilon$-suboptimal policy $\hat{\pi}$ for which $V^{\pi^*} - V^{\hat{\pi}} \leq \epsilon$ with total sample complexity of:*

$$n = \widetilde{O}\left(\frac{\max\{C_{\pi^*}^2, 1\} V_{\max}^2 dH^3 |\mathcal{A}| \log(HT|\mathcal{F}|/\delta)}{\epsilon^2}\right).$$

*Proof.* The following follows from a standard online-to-batch conversion. Setting $\pi^e = \pi^*$ in (15) and defining the policy $\hat{\pi} = \mathrm{Uniform}\left(\{\pi^1, \ldots, \pi^T\}\right)$, we get that

$$\mathbb{E}\left[V^{\pi^*} - V^{\hat{\pi}}\right] = \frac{1}{T}\left(\sum_{t=1}^{T} V^{\pi^*} - V^{\pi^t}\right) = O\left(\max\{C_{\pi^e}, 1\} V_{\max} \sqrt{\frac{dH^2}{T} \cdot \log\left(1 + \frac{T}{d}\right) \log\left(\frac{HT|\mathcal{F}|}{\delta}\right)}\right).$$

Thus, we the policy returned after $T \geq \widetilde{O}\left(\frac{\max\{C_{\pi^*}^2, 1\} V_{\max}^2 dH^2 \log(HT|\mathcal{F}|/\delta)}{\epsilon^2}\right)$ satisfies $\mathbb{E}\left[V^{\pi^*} - V^{\hat{\pi}}\right] \leq \epsilon$. In these $T$ iterations, the total number of offline samples used is

$$m_{\mathrm{off}} = T = \widetilde{O}\left(\frac{\max\{C_{\pi^*}^2, 1\} V_{\max}^2 dH^2 \log(HT|\mathcal{F}|/\delta)}{\epsilon^2}\right),$$

and the total number of online samples collected is

$$m_{\mathrm{on}} \cdot H \cdot T = \widetilde{O}\left(\frac{\max\{C_{\pi^*}^2, 1\} V_{\max}^2 dH^3 |\mathcal{A}| \log(HT|\mathcal{F}|/\delta)}{\epsilon^2}\right),$$

where the additional $H$ factor appears because we collect $m_{\mathrm{on}}$ samples for every $h \in [H]$ in the algorithm. $\square$

### A.4 LOW-RANK MDP

In this section, we briefly introduce the low-rank MDP model (Du et al., 2021), which is captured by the V-type Bilinear model discussed in Appendix A.3. Unlike the linear MDP model discussed in Section 5.1, low-rank MDP does not assume the feature $\phi$ is known a priori.

**Definition 5** (Low-rank MDP). *A MDP is called low-rank MDP if there exists $\mu^\star : \mathcal{S} \mapsto \mathbb{R}^d, \phi^\star : \mathcal{S} \times \mathcal{A} \mapsto \mathbb{R}^d$, such that the transition dynamics $P(s'|s,a) = \mu^\star(s')^\top \phi^\star(s,a)$ for all $s,a,s'$. We additionally assume that we are given a realizable representation class $\Phi$ such that $\phi^\star \in \Phi$, and that $\sup_{s,a} \|\phi^\star(s,a)\|_2 \le 1$, and $\|f^\top \mu^\star\|_2 \le \sqrt{d}$ for any $f : \mathcal{S} \mapsto [-1,1]$.*

Consider the function class $\mathcal{F}_h = \{w^\top \phi(s,a) : \phi \in \Phi, w \in \mathbb{B}_d(B_W)\}$, and through the bilinear decomposition we have that $B_W \le 2\sqrt{d}$. By inspection, we know that this function class satisfies Assumption 1. Furthermore, it is well known that the low rank MDP model has V-type bilinear rank of at most $d$ (Du et al., 2021). Invoking the sample complexity bound given in Corollary 2 for V-type Bilinear models, we get the following result.

**Lemma 10.** *Let $\delta \in (0,1)$ and $\Phi$ be a given representation class. Suppose that the MDP is a rank $d$ MDP w.r.t. some $\phi^\star \in \Phi$, $C_{\pi^*} < \infty$, and consider $\mathcal{F}_h$ defined above. Then, with probability $1 - \delta$, Algorithm 2 finds an $\epsilon$-suboptimal policy with total sample complexity (offline + online):*

$$\widetilde{O}\left( \frac{\max\{C_{\pi^*}^2, 1\} V_{\max}^2 d^2 H^4 |\mathcal{A}| \log(HTd|\Phi|/\epsilon\delta)}{\epsilon^2} \right).$$

*Proof sketch of Lemma 10.* The proof follows by invoking the result in Corollary 1 for a discretization of the class $\mathcal{F}$, denoted by $\mathcal{F}_\epsilon = \mathcal{F}_{0,\epsilon} \times \cdots \times \mathcal{F}_{H-1,\epsilon}$. $\mathcal{F}_\epsilon$ is defined such that $\mathcal{F}_{h,\epsilon} = \{w^\top \phi(s,a) : \phi \in \Phi, w \in \widehat{\mathbb{B}}_{d,\epsilon}(B_W)\}$ where $\widehat{\mathbb{B}}_{d,\epsilon}(B_W)$ is an $\epsilon$-net of the $\mathbb{B}_d(B_W)$ under $\ell_\infty$-distance and contains $O((B_W/\epsilon)^d)$ many elements. Thus, we get that $\log(|\mathcal{F}_\epsilon|) = O(Hd\log(B_W|\Phi|/\epsilon))$. $\square$

For low-rank MDP, the transfer coefficient $C_\pi$ is upper bounded by a relative condition number style quantity defined using the unknown ground truth feature $\phi^\star$ (see Lemma 13). On the computational side, Algorithm 1 (with the modification of $a \sim \text{Uniform}(\mathcal{A})$ in the online data collection step) requires to solve a least squares regression problem at every round. The objective of this regression problem is a convex functional of the hypothesis $f$ over the constraint set $\mathcal{F}$. While this is not fully efficiently implementable due to the potential non-convex constraint set $\mathcal{F}$ (e.g., $\phi$ could be complicated), our regression problem is still much simpler than the oracle models considered in the prior works for this model (Agarwal et al., 2020a; Sekhari et al., 2021; Uehara et al., 2021; Modi et al., 2021).

### A.5 BOUNDS ON TRANSFER COEFFICIENT

Note that $C_\pi$ takes both the distribution shift and the function class into consideration, and is smaller than the existing density ratio based concentrability coefficient (Kakade & Langford, 2002; Munos & Szepesvári, 2008; Chen & Jiang, 2019) and also existing Bellman error based concentrability coefficient Xie et al. (2021a). We formalize this in the following lemma.

**Lemma 11.** *For any $\pi$ and offline distribution $\nu$,*

$$C_\pi \le \sqrt{\max_{f,h} \frac{\|f_h - \mathcal{T}f_{h+1}\|_{d_h^\pi}^2}{\|f_h - \mathcal{T}f_{h+1}\|_{\nu_h}^2}} \le \sup_{h,s,a} \frac{d_h^\pi(s,a)}{\nu_h(s,a)}.$$

*Proof.* Using Jensen's inequality, we get that

$$C_\pi \le \sqrt{\max_f \frac{\sum_{h=0}^{H-1} \|f_h - \mathcal{T}f_{h+1}\|_{d_h^\pi}^2}{\sum_{h=0}^{H-1} \|f_h - \mathcal{T}f_{h+1}\|_{\nu_h}^2}}$$

$$\le \sqrt{\max_{f,h} \frac{\|f_h - \mathcal{T}f_{h+1}\|_{d_h^\pi}^2}{\|f_h - \mathcal{T}f_{h+1}\|_{\nu_h}^2}}$$

$$\leq \sqrt{\sup_{h,s,a} \frac{d_h^\pi(s,a)}{\nu_h(s,a)}}$$

$$\leq \sup_{h,s,a} \frac{d_h^\pi(s,a)}{\nu_h(s,a)},$$

where the second line follows from the Mediant inequality and the last line holds whenever $\sup_{h,s,a} \frac{d_h^\pi(s,a)}{\nu_h(s,a)} \geq 1$. $\qquad\square$

Next we show that in the linear Bellman complete setting, $C_\pi$ is bounded by the relative condition number using the linear features.

**Lemma 12.** *Consider the linear Bellman complete setting ([Definition 3](#)) with known feature $\phi$. Suppose that the feature covariance matrix induced by offline distribution $\nu$: $\Sigma_{\nu_h} := \mathbb{E}_{s,a\sim\nu_h}[\phi^\star(s,a)\phi^\star(s,a)^\top]$ is invertible. Then for any policy $\pi$, we have*

$$C_\pi \leq \sqrt{\max_h \mathbb{E}_{s,a\sim d_h^\pi} \|\phi(s,a)\|^2_{\Sigma_{\nu_h}^{-1}}}.$$

*Proof.* Repeating the argument in [Lemma 11](#), we have

$$C_\pi \leq \sqrt{\max_{f,h} \frac{\|f_h - \mathcal{T}f_{h+1}\|^2_{d_h^\pi}}{\|f_h - \mathcal{T}f_{h+1}\|^2_{\nu_h}}}$$

$$\leq \sqrt{\max_{w,h} \frac{\|w_h^\top \phi - w'^\top_h \phi\|^2_{d_h^\pi}}{\|w_h^\top \phi - w'^\top_h \phi\|^2_{\nu_h}}}$$

$$\leq \sqrt{\max_{w,h} \frac{\|(w_h - w'_h)\|^2_{\Sigma_{\nu_h}} \mathbb{E}_{d_h^\pi} \|\phi\|^2_{\Sigma_{\nu_h}^{-1}}}{\|(w_h - w'_h)^\top \phi\|^2_{\nu_h}}}$$

$$= \sqrt{\max_h \mathbb{E}_{s,a\sim d_h^\pi} \|\phi(s,a)\|^2_{\Sigma_{\nu_h}^{-1}}}.$$

Recall that in linear Bellman complete setting, we can write $f$ as $w^\top \phi$, and for any $w$ that defines $f$, there exists $w'$ such that $\mathcal{T}f = w'^\top \phi$. $\qquad\square$

Now we proceed to low-rank MDPs where feature is unknown. We show that for low-rank MDPs, $C_\pi$ is bounded by the partial feature coverage using the unknown ground truth feature.

**Lemma 13.** *Consider the low-rank MDP setting ([Definition 5](#)) where the transition dynamics $P$ is given by $P(s' \mid s,a) = \langle \mu^\star(s'), \phi^\star(s,a) \rangle$ for some $\mu^\star, \phi^\star \in \mathbb{R}^d$. Suppose that the offline distribution $\nu = (\nu_0, \ldots, \nu_{H-1})$ is such that $\max_h \max_{s,a} \frac{\pi_h(a|s)}{\nu_h(a|s)} \leq \alpha$ for any $s,a$. Furthermore, suppose that $\nu$ is induced via trajectories i.e. $\nu_0(s) = d_0(s)$ and $\nu_h(s) = \mathbb{E}_{\bar{s},\bar{a}\sim\nu_{h-1}} P(s|\bar{s},\bar{a})$ for any $h \geq 1$, and that the feature covariance matrix $\Sigma_{\nu_{h-1},\phi^\star} := \mathbb{E}_{s,a\sim\nu_{h-1}}[\phi^\star(s,a)\phi^\star(s,a)^\top]$ is invertible.[6] Then for any policy $\pi$, we have*

$$C_\pi \leq \sqrt{\alpha} \sum_{h=1}^{H} \mathbb{E}_{s,a\sim d_{h-1}^\pi}\left[\|\phi^\star(s,a)\|_{\Sigma_{\nu_{h-1},\phi^\star}^{-1}}\right] + \sqrt{\alpha}.$$

*Proof.* We first upper bound the numerator separately. First note that for $h = 0$,

$$\mathbb{E}_{s,a\sim d_0^\pi}\left[\mathcal{T}f_1(s,a) - f_0(s,a)\right] \leq \sqrt{\mathbb{E}_{s\sim d_0, a\sim\pi(\cdot|s)}\left[(\mathcal{T}f_1(s,a) - f_0(s,a))^2\right]}$$

$$\leq \sqrt{\max_{s,a} \frac{d_0^\pi(s,a)}{\nu_0(s,a)} \cdot \mathbb{E}_{s,a\sim\nu_0}\left[(\mathcal{T}f_1(s,a) - f_0(s,a))^2\right]}$$

---

[6]This is for notation simplicity, and we emphasize that we do not assume eigenvalues are lower bounded. In other words, eigenvalue of this feature covariance matrix could approach to $0^+$.

$$\leq \sqrt{\alpha \cdot \mathbb{E}_{s,a\sim\nu_0}\left[\left(\mathcal{T}f_1(s,a) - f_0(s,a)\right)^2\right]}, \tag{16}$$

where the last inequality follows from our assumption since $\max_{s,a} \frac{d_0^\pi(s,a)}{\nu_0(s,a)} = \max_{s,a} \frac{\pi_0(a|s)}{\nu_0(a|s)} \leq \alpha$.

Next, for any $h \geq 1$, we note that backing up one step and looking at the pair $\bar{s}, \bar{a}$ that lead to the state $s$, we get that

$$
\begin{aligned}
&\mathbb{E}_{s,a\sim d_h^\pi}\left[\mathcal{T}f_{h+1}(s,a) - f_h(s,a)\right] \\
&= \mathbb{E}_{\bar{s},\bar{a}\sim d_{h-1}^\pi, s\sim P(\bar{s},\bar{a}), a\sim\pi(s)}\left[\mathcal{T}f_{h+1}(s,a) - f_h(s,a)\right] \\
&= \mathbb{E}_{\bar{s},\bar{a}\sim d_{h-1}^\pi}\left[\int \left(\phi^\star(\bar{s},\bar{a})^\top \mu^\star(s)\right) \sum_a \pi(a|s)\left[\mathcal{T}f_{h+1}(s,a) - f_h(s,a)\right]\mathrm{d}s\right] \\
&= \mathbb{E}_{\bar{s},\bar{a}\sim d_{h-1}^\pi}\left[\phi^\star(\bar{s},\bar{a})^\top \int \sum_a \mu^\star(s)\pi(a|s)\left[\mathcal{T}f_{h+1}(s,a) - f_h(s,a)\right]\mathrm{d}s\right] \\
&\leq \mathbb{E}_{\bar{s},\bar{a}\sim d_{h-1}^\pi}\left[\|\phi^\star(\bar{s},\bar{a})\|_{\Sigma_{\nu_{h-1},\phi^\star}^{-1}} \left\|\int \sum_a \mu^\star(s)\pi(a|s)\left[\mathcal{T}f_{h+1}(s,a) - f_h(s,a)\right]\mathrm{d}s\right\|_{\Sigma_{\nu_{h-1},\phi^\star}}\right],
\end{aligned}
\tag{17}
$$

where the last line follows from an application of Cauchy-Schwarz inequality. For the term inside the expectation in the right hand side above, we note that,

$$
\begin{aligned}
&\left\|\int \sum_a \mu^\star(s)\pi(a|s)\left[\mathcal{T}f_{h+1}(s,a) - f_h(s,a)\right]\mathrm{d}s\right\|_{\Sigma_{\nu_{h-1},\phi^\star}}^2 \\
&\overset{(i)}{=} \mathbb{E}_{\bar{s},\bar{a}\sim\nu_{h-1}}\left[\left(\int \sum_a \left(\mu^\star(s)^\top \phi^\star(\bar{s},\bar{a})\right)\pi(a|s)(\mathcal{T}f_{h+1}(s,a) - f_h(s,a))\mathrm{d}s\right)^2\right] \\
&= \mathbb{E}_{\bar{s},\bar{a}\sim\nu_{h-1}}\left[\left(\mathbb{E}_{s\sim P(\bar{s},\bar{a}), a\sim\pi(s)}[\mathcal{T}f_{h+1}(s,a) - f_h(s,a)]\right)^2\right] \\
&\overset{(ii)}{\leq} \mathbb{E}_{\bar{s},\bar{a}\sim\nu_{h-1}, s\sim P(\bar{s},\bar{a}), a\sim\pi(s)}\left[(\mathcal{T}f_{h+1}(s,a) - f_h(s,a))^2\right] \\
&\overset{(iii)}{=} \mathbb{E}_{s\sim\nu_h, a\sim\pi(s)}\left[(\mathcal{T}f_{h+1}(s,a) - f_h(s,a))^2\right] \\
&\overset{(iv)}{\leq} \alpha \cdot \mathbb{E}_{s,a\sim\nu_h}\left[(\mathcal{T}f_{h+1}(s,a) - f_h(s,a))^2\right]
\end{aligned}
\tag{18}
$$

where $(i)$ follows by expanding the norm , $(ii)$ follows an application of Jensen's inequality, $(iii)$ is due to our assumption that the offline dataset is generated using trajectories such that $\nu_h(s) = \mathbb{E}_{\bar{s},\bar{s}\sim\nu_{h-1}}[P(s \mid \bar{s},\bar{a})]$. Finally, $(iv)$ follows from the definition of $\alpha$. Plugging (18) in (17), we get that for $h \geq 1$,

$$
\begin{aligned}
&\mathbb{E}_{s,a\sim d_h^\pi}\left[\mathcal{T}f_{h+1}(s,a) - f_h(s,a)\right] \\
&\leq \mathbb{E}_{\bar{s},\bar{a}\sim d_{h-1}^\pi}\left[\|\phi^\star(\bar{s},\bar{a})\|_{\Sigma_{\nu_{h-1},\phi^\star}^{-1}} \sqrt{\alpha \cdot \mathbb{E}_{s,a\sim\nu_h}\left[(\mathcal{T}f_{h+1}(s,a) - f_h(s,a))^2\right]}\right]
\end{aligned}
\tag{19}
$$

We are now ready to bound the transfer coefficient. First note that using (16), for any $f$,

$$
\frac{\mathbb{E}_{s,a\sim d_0^\pi}\left[\mathcal{T}f_1(s,a) - f_0(s,a)\right]}{\sqrt{\sum_{h=0}^{H-1}\mathbb{E}_{s,a\sim\nu_h}\left[(\mathcal{T}f_{h+1}(s,a) - f_h(s,a))^2\right]}} \leq \frac{\sqrt{\alpha \cdot \mathbb{E}_{s,a\sim\nu_0}\left[(\mathcal{T}f_1(s,a) - f_0(s,a))^2\right]}}{\sqrt{\sum_{h=0}^{H-1}\mathbb{E}_{s,a\sim\nu_h}\left[(\mathcal{T}f_{h+1}(s,a) - f_h(s,a))^2\right]}}
$$

$$\leq \sqrt{\alpha}.$$

Furthermore, for any $f$, using (19), we get that

$$\frac{\sum_{h=1}^{H-1} \mathbb{E}_{s,a\sim d_h^\pi}[\mathcal{T}f_{h+1}(s,a) - f_h(s,a)]}{\sqrt{\sum_{h=0}^{H-1} \mathbb{E}_{s,a\sim\nu_h}\Big[(\mathcal{T}f_{h+1}(s,a) - f_h(s,a))^2\Big]}}$$

$$\leq \sum_{h=1}^{H-1} \mathbb{E}_{\bar{s},\bar{a}\sim d_{h-1}^\pi}\left[\|\phi^\star(\bar{s},\bar{a})\|_{\Sigma_{\nu_{h-1},\phi^\star}^{-1}} \frac{\sqrt{\alpha\cdot\mathbb{E}_{s,a\sim\nu_h}\Big[(\mathcal{T}f_{h+1}(s,a) - f_h(s,a))^2\Big]}}{\sqrt{\sum_{h=0}^{H-1}\mathbb{E}_{s,a\sim\nu_h}\Big[(\mathcal{T}f_{h+1}(s,a) - f_h(s,a))^2\Big]}}\right]$$

$$\leq \sum_{h=1}^{H} \mathbb{E}_{\bar{s},\bar{a}\sim d_{h-1}^\pi}\left[\|\phi^\star(\bar{s},\bar{a})\|_{\Sigma_{\nu_{h-1},\phi^\star}^{-1}}\sqrt{\alpha}\right],$$

where the last line holds for an appropriate choice of $\lambda$ (e.g. $\lambda = 0$). Combining the above two bounds in the definition of $C_\pi$ we get that

$$C_\pi \leq \sqrt{\alpha}\sum_{h=1}^{H}\mathbb{E}_{\bar{s},\bar{a}\sim d_{h-1}^\pi}\left[\|\phi^\star(\bar{s},\bar{a})\|_{\Sigma_{\nu_{h-1},\phi^\star}^{-1}}\right] + \sqrt{\alpha}.$$

$\square$

Note that in the above result, the transfer coefficient is upper bounded by the relative coverage under unknown feature $\phi^\star$ and a term $\alpha$ related to the action coverage, i.e., $\max_h \max_{s,a} \frac{\pi_h(a|s)}{\nu_h(a|s)} \leq \alpha$. This matches to the coverage condition used in prior offline RL works for low-rank MDPs (Uehara & Sun, 2021).

## B   AUXILIARY LEMMAS

In this section, we provide a few results and their proofs that we used in the previous sections. We first with the following form of Freedman's inequality that is a modification of a similar inequality in (Beygelzimer et al., 2011).

**Lemma 14** (Freedman's Inequality). *Let $\{X_1,\ldots,X_T\}$ be a sequence of non-negative random variables where each $x_t$ is sampled from some process that depends on all previous instances, i.e, $x_t \sim \rho_t = \rho_t(x_{1:t-1})$. Further, suppose that $|X_t| \leq R$ almost surely for all $t \leq T$. Then, for any $\delta > 0$ and $\lambda \in [0, 1/2R]$, with probability at least $1 - \delta$,*

$$\left|\sum_{t=1}^{T} X_t - \mathbb{E}[X_t \mid \rho_t]\right| \leq \lambda \sum_{t=1}^{T}\big(2R|\mathbb{E}[X_t \mid \rho_t]| + \mathbb{E}\big[X_t^2 \mid \rho_t\big]\big) + \frac{\log(2/\delta)}{\lambda}.$$

*Proof.* Define the random variable $Z_t = X_t - \mathbb{E}[X_t \mid \rho_t]$. Clearly, $\{Z_t\}_{t=1}^{T}$ is a martingale difference sequence. Furthermore, we have that for any $t$, $|Z_t| \leq 2R$ and that

$$\mathbb{E}\big[Z_t^2 \mid \rho_t\big] = \mathbb{E}\Big[(X_t - \mathbb{E}[X_t \mid \rho_t])^2 \mid \rho_t\Big] \leq 2R|\mathbb{E}[X_t \mid \rho_t]| + \mathbb{E}\big[X_t^2 \mid \rho_t\big]. \qquad (20)$$

where the last inequality holds because $|X_t| \leq R$.

Using the form of Freedman's inequality in Beygelzimer et al. (2011, Theorem 1), we get that for any $\lambda \in [0, 1/2R]$,

$$\left|\sum_{t=1}^{T} Z_t\right| \leq \lambda \sum_{t=1}^{T}\mathbb{E}\big[Z_t^2 \mid \rho_t\big] + \frac{\log(2/\delta)}{\lambda}.$$

Plugging in the form of $Z_t$ and using (20), we get the desired statement. $\square$

Next we give a formal proof of Lemma 2, which gives a generalization bound for least squares regression when the samples are adapted to an increasing filtration (and are not necessarily i.i.d.). The proof follows similarly to Agarwal et al. (2019, Lemma A.11).

**Lemma 15** (Lemma 2 restated: Least squares generalization bound). *Let $R > 0$, $\delta \in (0, 1)$, we consider a sequential function estimation setting, with an instance space $\mathcal{X}$ and target space $\mathcal{Y}$. Let $\mathcal{H} : \mathcal{X} \mapsto [-R, R]$ be a class of real valued functions. Let $\mathcal{D} = \{(x_1, y_1), \ldots, (x_T, y_T)\}$ be a dataset of $T$ points where $x_t \sim \rho_t = \rho_t(x_{1:t-1}, y_{1:t-1})$, and $y_t$ is sampled via the conditional probability $p(\cdot \mid x_t)$:*

$$y_t \sim p(\cdot \mid x_t) := h^*(x_t) + \varepsilon_t,$$

*where the function $h^*$ satisfies approximate realizability i.e.*

$$\inf_{h \in \mathcal{H}} \frac{1}{T} \sum_{t=1}^{T} \mathbb{E}_{x \sim \rho_t} \left[ (h^*(x) - h(x))^2 \right] \leq \gamma,$$

*and $\{\epsilon_i\}_{i=1}^n$ are independent random variables such that $\mathbb{E}[y_t \mid x_t] = h^*(x_t)$. Additionally, suppose that $\max_t |y_t| \leq R$ and $\max_x |h^*(x)| \leq R$. Then the least square solution $\widehat{h} \leftarrow \operatorname{argmin}_{h \in \mathcal{H}} \sum_{t=1}^{T} (h(x_t) - y_t)^2$ satisfies with probability at least $1 - \delta$,*

$$\sum_{t=1}^{T} \mathbb{E}_{x \sim \rho_t} \left[ (\widehat{h}(x) - h^*(x))^2 \right] \leq 3\gamma T + 256 R^2 \log(2|\mathcal{H}|/\delta).$$

*Proof.* Consider any fixed function $h \in \mathcal{H}$ and define the random variable

$$Z_t^h := (h(x_t) - y_t)^2 - (h^*(x_t) - y_t)^2.$$

Define the notation $\mathbb{E}[\cdot \mid \rho_t]$ to denote $\mathbb{E}_{x_t \sim \rho_t}[\cdot]$, and note that

$$\mathbb{E}\left[ Z_t^h \mid \rho_t \right] = \mathbb{E}_{x_t \sim \rho_t}[(h(x_t) - h^*(x_t))(h(x_t) + h^*(x_t) - 2y_i)] = \mathbb{E}_{x_t \sim \rho_t} \left[ (h(x_t) - h^*(x_t))^2 \right], \tag{21}$$

where the last line holds because $\mathbb{E}[y_t \mid x_t] = h^*(x_t)$. Furthermore, we also have that

$$\mathbb{E}\left[ (Z_t^h)^2 \mid \rho_t \right] = \mathbb{E}_{x_t \sim \rho_t} \left[ (h(x_t) - h^*(x_t))^2 (h(x_t) + h^*(x_t) - 2y_t)^2 \right]$$
$$\leq 16 R^2 \mathbb{E}_{x_t \sim \rho_t} \left[ (h(x_t) - h^*(x_t))^2 \right]. \tag{22}$$

Now we can note that the sequence of random variables $\{Z_1^h, \ldots, Z_T^h\}$ satisfies the condition in Lemma 14 with $|Z_t^h| \leq 4R^2$. Thus we get that for any $\lambda \in [0, 1/8R^2]$ and $\delta > 0$, with probability at least $1 - \delta$,

$$\left| \sum_{t=1}^{T} Z_t^h - \mathbb{E}\left[ Z_t^h \mid \rho_t \right] \right| \leq \lambda \sum_{t=1}^{T} \left( 8R^2 \left| \mathbb{E}\left[ Z_t^h \mid \rho_t \right] \right| + \mathbb{E}\left[ (Z_t^h)^2 \mid \rho_t \right] \right) + \frac{\log(2/\delta)}{\lambda}$$
$$\leq 32\lambda R^2 \sum_{t=1}^{T} \mathbb{E}_{x_t \sim \rho_t} \left[ (h(x_t) - h^*(x_t))^2 \right] + \frac{\log(2/\delta)}{\lambda},$$

where the last inequality uses (21) and (22). Setting $\lambda = 1/64R^2$ in the above, and taking a union bound over $h$, we get that for any $h \in \mathcal{H}$ and $\delta > 0$, with probability at least $1 - \delta$,

$$\left| \sum_{t=1}^{T} Z_t^h - \mathbb{E}\left[ Z_t^h \mid \rho_t \right] \right| \leq \frac{1}{2} \sum_{t=1}^{T} \mathbb{E}_{x_t \sim \rho_t} \left[ (h(x_t) - h^*(x_t))^2 \right] + 64 R^2 \log(2|\mathcal{H}|/\delta).$$

Rearranging the terms and using (21) in the above implies that,

$$\sum_{t=1}^{T} Z_t^h \leq \frac{3}{2} \sum_{t=1}^{T} \mathbb{E}_{x_t \sim \rho_t} \left[ (h(x_t) - h^*(x_t))^2 \right] + 64 R^2 \log(2|\mathcal{H}|/\delta)$$

and

$$\sum_{t=1}^{T} \mathbb{E}_{x_t \sim \rho_t} \left[ (h(x_t) - h^*(x_t))^2 \right] \leq 2 \sum_{t=1}^{T} Z_t^h + 128 R^2 \log(2|\mathcal{H}|/\delta). \tag{23}$$

For the rest of the proof, we condition on the event that (23) holds for all $h \in \mathcal{H}$.

Define the function $\widetilde{h} := \operatorname{argmin}_{h \in \mathcal{H}} \sum_{t=1}^{T} \mathbb{E}_{x_t \sim \rho_t} [(h(x_t) - h^*(x_t))^2]$. Using (23), we get that

$$\sum_{t=1}^{T} Z_t^{\widetilde{h}} \leq \frac{3}{2} \sum_{t=1}^{T} \mathbb{E}_{x_t \sim \rho_t} \left[ (\widetilde{h}(x_t) - h^*(x_t))^2 \right] + 64R^2 \log(2|\mathcal{H}|/\delta)$$

$$\leq \frac{3}{2} \gamma T + 64R^2 \log(2|\mathcal{H}|/\delta),$$

where the last inequality follows from the approximate realizability assumption. Let $\widehat{h}$ denote the least squares solution on dataset $\{(x_t, y_t)\}_{t \leq T}$. By definition, we have that

$$\sum_{t=1}^{T} Z_t^{\widehat{h}} = (\widehat{h}(x_t) - y_t)^2 - (h^*(x_t) - y_t)^2 \leq (\widetilde{h}(x_t) - y_t)^2 - (h^*(x_t) - y_t)^2 = \sum_{t=1}^{T} Z_t^{\widetilde{h}}.$$

Combining the above two relations, we get that

$$\sum_{t=1}^{T} Z_t^{\widehat{h}} \leq \frac{3}{2} \gamma T + 64R^2 \log(2|\mathcal{H}|/\delta). \tag{24}$$

Finally, using (23) for the function $\widehat{h}$, we get that

$$\sum_{t=1}^{T} \mathbb{E}_{x_t \sim \rho_t} \left[ (\widehat{h}(x_t) - h^*(x_t))^2 \right] \leq 2 \sum_{t=1}^{T} Z_t^{\widehat{h}} + 128R^2 \log(2|\mathcal{H}|/\delta)$$

$$\leq 3\gamma T + 256R^2 \log(2|\mathcal{H}|/\delta),$$

where the last inequality uses the relation (24). □

## C  Low Bellman Eluder Dimension problems

In this section, we consider problems with low Bellman Eluder dimensions Jin et al. (2021a). This complexity measure is a distributional version of the Eluder dimension applied to the class of Bellman residuals w.r.t. $\mathcal{F}$. We show that our algorithm Hy-Q gives a similar performance guarantee for problems with small Bellman Eluder dimensions. This demonstrates that Hy-Q applies to any general model-free RL frameworks known in the RL literature so far.

We first introduce the key definitions:

**Definition 6** ($\varepsilon$-independence between distributions (Jin et al., 2021a)). *Let $\mathcal{G}$ be a class of functions defined on a space $\mathcal{X}$, and $\nu, \mu_1, \ldots, \mu_n$ be probability measures over $\mathcal{X}$. We say $\nu$ is $\varepsilon$-independent of $\{\mu_1, \mu_2, \ldots, \mu_n\}$ with respect to $\mathcal{G}$ if there exists $g \in \mathcal{G}$ such that $\sqrt{\sum_{i=1}^{n} (\mathbb{E}_{\mu_i}[g])^2} \leq \varepsilon$, but $|\mathbb{E}_\nu[g]| > \varepsilon$.*

**Definition 7** (Distributional Eluder (DE) dimension). *Let $\mathcal{G}$ be a function class defined on $\mathcal{X}$, and $\mathcal{P}$ be a family of probability measures over $\mathcal{X}$. The distributional Eluder dimension $\dim_{\mathrm{DE}}(\mathcal{F}, \mathcal{P}, \varepsilon)$ is the length of the longest sequence $\{\rho_1, \ldots, \rho_n\} \subset \mathcal{P}$ such that there exists $\varepsilon' \geq \varepsilon$ where $\rho_i$ is $\varepsilon'$-independent of $\{\rho_1, \ldots, \rho_{i-1}\}$ for all $i \in [n]$.*

**Definition 8** (Bellman Eluder (BE) dimension (Jin et al., 2021a)). *Given a value function class $\mathcal{F}$, let $\mathcal{G}_h := (f_h - \mathcal{T} f_{h+1} \mid f \in \mathcal{F}_h, f_{h+1} \in \mathcal{F}_{h+1})$ be the set of Bellman residuals induced by $\mathcal{F}$ at step $h$, and $\mathcal{P} = \{\mathcal{P}_h\}_{h=1}^{H}$ be a collection of $H$ probability measure families over $\mathcal{X} \times \mathcal{A}$. The $\epsilon$-Bellman Eluder dimension of $\mathcal{F}$ with respect to $\mathcal{P}$ is defined as*

$$\dim_{\mathrm{BE}}(\mathcal{F}, \mathcal{P}, \varepsilon) := \max_{h \in [H]} \dim_{\mathrm{DE}}(\mathcal{G}_h, \mathcal{P}_h, \epsilon).$$

We also note the following lemma that controls the rate at which Bellman error accumulates.

**Lemma 16** (Lemma 41, (Jin et al., 2021a)). *Given a function class $\mathcal{G}$ defined on a space $\mathcal{X}$ with $\sup_{g \in \mathcal{G}, x \in \mathcal{X}} |g(x)| \leq C$, and a set of probability measures $\mathcal{P}$ over $\mathcal{X}$. Suppose that the sequence*

$\{g_k\}_{k=1}^K \subset \mathcal{G}$ and $\{\mu_k\}_{k=1}^K \subset \mathcal{P}$ satisfy that $\sum_{t=1}^{k-1} (\mathbb{E}_{\mu_t}[g_k])^2 \leq \beta$ for all $k \in [K]$. Then, for all $k \in [K]$ and $\gamma > 0$,

$$\sum_{t=1}^k |\mathbb{E}_{\mu_t}[g_t]| \leq O\Big(\sqrt{\dim_{\mathrm{DE}}(\mathcal{G}, \mathcal{P}, \gamma)\beta k} + \min\{k, \dim_{\mathrm{DE}}(\mathcal{G}, \mathcal{P}, \gamma)C\} + k\gamma\Big).$$

We next state our main theorem whose proof is similar to that of Theorem 1.

**Theorem 3** (Cumulative suboptimality). *Fix $\delta \in (0, 1)$, $m_{\mathrm{off}} = HT/d$ and $m_{\mathrm{on}} = H^2$, and suppose that the underlying MDP admits Bellman eluder dimention $d$, and the function class $\mathcal{F}$ satisfies Assumption 1. Then with probability at least $1 - \delta$, Algorithm 1 obtains the following bound on cumulative subpotimality w.r.t. any comparator policy $\pi^e$,*

$$\sum_{t=1}^T V^{\pi^e} - V^{\pi^t} = \widetilde{O}\Big(V_{\max} \max\{C_{\pi^e}, 1\}\sqrt{dT \cdot \log(H|\mathcal{F}|/\delta)}\Big),$$

*where $\pi^t = \pi^{f^t}$ is the greedy policy w.r.t. $f^t$ at round $t$ and $d = \dim_{\mathrm{BE}}(\mathcal{F}, \mathcal{P}_{\mathcal{F}}, 1/\sqrt{T})$. Here $\mathcal{P}_{\mathcal{F}}$ is the class of occupancy measures that can be be induced by greedy policies w.r.t. value functions in $\mathcal{F}$.*

*Proof.* Repeating the analysis till (10) in the proof of Theorem 1, we get that

$$\sum_{t=1}^T V^{\pi^e} - V^{\pi^t} \leq TC_{\pi^e} \cdot \sqrt{H \cdot \Delta_{\mathrm{off}}} + \sum_{t=1}^T \sum_{h=0}^{H-1} \Big|\mathbb{E}_{s,a \sim d_h^{\pi^{f^t}}} \big[f_h^t(s, a) - \mathcal{T}_h f_{h+1}^t(s, a)\big]\Big|$$

Using the bound in Lemma 6 and Lemma 16 in the above, we get that

$$\sum_{t=1}^T V^{\pi^e} - V^{\pi^t} \lesssim TC_{\pi^e} \cdot \sqrt{H \cdot \Delta_{\mathrm{off}}} + \sum_{h=0}^{H-1} \sqrt{\dim_{\mathrm{DE}}(\mathcal{G}_h, \mathcal{P}_{\mathcal{F};h}, \gamma)\Delta_{\mathrm{on}}T}$$
$$+ \min\{T, \dim_{\mathrm{DE}}(\mathcal{G}_h, \mathcal{P}_{\mathcal{F};h}, \gamma)C\} + T\gamma.$$

where $\mathcal{G}_h := (f_h - \mathcal{T}f_{h+1} \mid f \in \mathcal{F}_h, f_{h+1} \in \mathcal{F}_{h+1})$ denotes the set of Bellman residuals induced by $\mathcal{F}$ at step $h$, and $\mathcal{P} = \{\mathcal{P}_{\mathcal{F};h}\}_{h=1}^H$ is the collection of occupancy measures at step $h$ induced by greedy policies w.r.t. value functions in $\mathcal{F}$. We set $\gamma = 1/\sqrt{T}$ and define $d = \dim_{\mathrm{BE}}(\mathcal{F}, \mathcal{P}, \gamma) = \max_h \dim_{\mathrm{DE}}(\mathcal{G}_h, \mathcal{P}_{\mathcal{F};h}, \gamma)$. Ignoring the lower order terms, we get that

$$\sum_{t=1}^T V^{\pi^e} - V^{\pi^t} \lesssim TC_{\pi^e} \cdot \sqrt{H \cdot \Delta_{\mathrm{off}}} + H\sqrt{d\Delta_{\mathrm{on}}T}$$
$$\lesssim TC_{\pi^e}V_{\max} \cdot \sqrt{H \cdot \frac{\log(HT|\mathcal{F}|/\delta)}{m_{\mathrm{off}}}} + HV_{\max}\sqrt{dT \cdot \frac{\log(HT|\mathcal{F}|/\delta)}{m_{\mathrm{on}}}},$$

where $\lesssim$ hides lower order terms, multiplying constants and log factors. Setting $m_{\mathrm{off}} = HT/d$ and $m_{\mathrm{on}} = H^2$, we get that

$$\sum_{t=1}^T V^{\pi^e} - V^{\pi^t} = \widetilde{O}\Big(C_{\pi^e}V_{\max}\sqrt{dT\log(HT|\mathcal{F}|/\delta)}\Big).$$

$\square$

## D  COMPARISON WITH PREVIOUS WORKS

As mentioned in the main text, many previous empirical works consider combining offline expert demonstrations with online interaction (Rajeswaran et al., 2017; Hester et al., 2018; Nair et al., 2018; 2020; Vecerik et al., 2017; Lee et al., 2022; Jia et al., 2022; Niu et al., 2022). Thus the idea of performing RL algorithm on both offline data (expert demonstrations) and online data is also explored in some of the previous works, for example, Vecerik et al. (2017) runs DDPG on both the online

and expert data, and Hester et al. (2018) uses DQN on both data but with an additional supervised loss. Since we already compared with Hester et al. (2018) in the experiment, here we focus on our discussion with Vecerik et al. (2017).

We first emphasize that Vecerik et al. (2017) only focuses on expert demonstrations and their experiments entirely rely on using expert demonstrations, while we focus on more general offline dataset that is not necessarily coming from experts. Said though, the DDPG-based algorithm from Vecerik et al. (2017) potentially can be used when offline data is not from experts. Although the algorithm from Vecerik et al. (2017) and Hy-Q share the same high-level intuition that one should perform RL on both the datasets, there are still a few differences : (1) Hy-Q uses Q-learning instead of deterministic policy gradients; note that deterministic policy gradient methods cannot be directly applied to discrete action setting; (2) Hy-Q does not require n-step TD style update, since in off-policy case, without proper importance weighting, n-step TD could incur strong bias. While proper tuning on n could balance bias and variance, one does not need to tune such n-step at all in Hy-Q; (3) The idea of keeping a non-zero ratio to sample offline dataset is also proposed in Vecerik et al. (2017). Our buffer ratio is derived from our theory analysis but meanwhile proves the advantage of the similar heuristic applied in Vecerik et al. (2017). (4) In their experiment, Vecerik et al. (2017) only considers expert demonstrations. In our experiment, we considered offline datasets with different amounts of transitions from very low-quality policies and showed Hy-Q is robust to low-quality transitions in offline data. Note that some of the differences may seem minor on the implementation level, but they may be important to the theory.

Regarding the experiments, our experimental evaluation adds the following insights over those in Vecerik et al. (2017): (i) hybrid methods can succeed without expert data, (ii) hybrid methods can succeed in hard exploration discrete-action tasks, (iii) the core algorithm ($Q$-learning vs DDPG) is not essential although some details may matter. Due to the similarity between the two methods, we believe some of these insights may also translate to Vecerik et al. (2017) and we expect that the choice between Hy-Q and Hy-DDPG will be environment specific, as it is with the purely online versions of these methods. In some situations, $Q$-learning works does not immediately imply Deterministic policy gradient methods work, nor vice versa. Nevertheless, it is beyond the scope of this paper to rigorously verify this claim and we deem the study of Actor-critic algorithms in Hybrid RL setting an interesting future direction.

## E   EXPERIMENT DETAILS

### E.1   COMBINATION LOCK

In this section we provide a detailed description of combination lock experiment. The combination lock environment has a horizon $H$ and 10 actions at each state. There are three latent states $z_{i,h}, i \in \{0, 1, 2\}$ for each timestep $h$, where $z_{i,h}, i \in \{0, 1\}$ are good states and $z_{2,h}$ is the bad state. For each good state, we randomly pick a good action $a_{i,h}$, such that in latent state $z_{i,h}, i \in \{0, 1\}$, taking the good action $a_{i,h}$ will result in 0.5 probability of transiting to $z_{0,h+1}$ and 0.5 probability of transiting to $z_{1,h+1}$ while taking all other actions will result in a 1 probability of transiting to $z_{2,h+1}$. At $z_{2,h}$, all actions will result in a deterministic transition to $z_{2,h+1}$. For the reward, we give an optimal reward of 1 for landing $z_{i,H}, i \in \{0, 1\}$. We also give an anti-shaped reward of 0.1 for all transitions from a good state to a bad state. All other transitions have a reward of 0. The initial distribution is a uniform distribution over $z_{0,0}$ and $z_{1,0}$. The observation space has dimension $2^{\lceil \log(H+1) \rceil}$, created by concatenating a one-hot representation of the latent state and a one-hot representation of the horizon (appending 0 if necessary). Random noise from $\mathcal{N}(0, 0.1)$ is added to each dimension, and finally the observation is multiplied by a Hadamard matrix. Note that in this environment, the agent needs to perform optimally for all $H$ timesteps to hit the final good state for an optimal reward of 1. Once the agent chooses a bad action, it will stay in the bad state until the end with at most 0.1 possible reward for the trajectory received while transitting from a good state to a bad state.

### E.2   IMPLEMENTATION DETAILS OF COMBINATION LOCK EXPERIMENT

We train $H$ separate Q-functions for all $H$ timesteps. Our function class consists of an encoder and a decoder. For the encoder, we feed the observation into one linear layer with 3 outputs, followed by

a softmax layer to get a state-representation. This design of encoder is intended to learn a one-hot representation of the latent state. We take a Kronecker Product of the state-representation and the action, and feed the result to a linear layer with only one output, which will be our Q value. In order to stabilize the training, we warm-start the Q-function of timestep $h - 1$ with the encoder from $h$ Q-function of the current iteration and the decoder from the $h - 1$ Q-function of the last iteration, for each iteration of training.

One remark is that since combination lock belongs to Block MDPs, we require a V-type algorithm instead of the Q-type algorithm as shown in the main text. The only difference lies in the online sampling process: instead of sampling from $d_h^{\pi^t}$, for each $h$, we sample from $d_h^{\pi^t} \circ \mathrm{Uniform}(\mathcal{A})$, i.e., we first rollin with respect to $\pi^t$ to timestep $h - 1$, then take a random action, observe the transition and collect that tuple. We provide Algorithm 2 for completeness. Note that the only difference is in line 4.

For CQL, we implemented the variant of CQL-DQN and picked the peak in the learning curve to report in the main paper (so it should represent an upper bound of the performance of CQL).

### E.3 IMPLEMENTATION DETAILS OF MONTEZUMA'S REVENGE EXPERIMENT

In this section we provide the detailed algorithm for the discounted setting. The overall algorithm is described in Algorithm 3. For the function approximation, we use a class of convolutional neural networks (parameterized by class $\Theta$) as promoted by the original DQN paper. We include several standard empirical design choices that have been practically proven to stabilize the training: we use Prioritize Experience Replay (Schaul et al., 2015) for our buffer. We also add Double DQN (Van Hasselt et al., 2016) and Dueling DQN (Wang et al., 2016) during our Q-update. We also observe that a decaying schedule on the offline sample ratio $\beta$ and the exploration rate $\epsilon$ also helps provide better performance. Note that an annealing $\beta$ does not contradict to our comment in Section 4 on catastrophic forgetting because we set $\beta$ to small after our online trajectory distribution covers $d^{\pi^e}$. In addition, we also perform per step update instead of per episode update since this has been the popular design choice and leads to better efficiency in practice.

### E.4 BASELINE IMPLEMENTATION

#### E.4.1 COMBINATION LOCK

We use the open-sourced implementation `https://github.com/BY571/CQL/tree/main/CQL-DQN` for CQL. For BRIEE, we use the official code released by the authors: `https://github.com/yudasong/briee`, where we rely on the code there for the combination lock environment.

#### E.4.2 MONTEZUMA'S REVENGE

We use the open-sourced implementation `https://github.com/jcwleo/random-network-distillation-pytorch` for RND. For CQL, we use `https://github.com/takuseno/d3rlpy` for their implementation of CQL for atari. We use `https://github.com/felix-kerkhoff/DQfD` for DQFD. For all baselines, we keep the hyperparameters used in these public repositories. For CQL and DQFD, we provide the offline datasets as described in the main text instead of using the offline dataset provided in the public repositories.[7] All baselines are tested in the same stochastic environment setup as in Burda et al. (2018).

### E.5 HARDWARE INFRASTRUCTURE

We run our experiments on a cluster of computes with Nvidia RTX 3090 GPUs and various CPUs which do not incur any randomness to the results.

---

[7]We note that CQL also fails completely with the original offline dataset (with 1 million samples) provided in the public repository.

---

**Algorithm 3** Discounted Hy-Q

---

**Require:** Value function class: $\mathcal{F}$ (induced by $\Theta$), #iterations: $T$, Offline dataset $\mathcal{D}^\nu$ of size $m_{\text{off}}$, discounted factor $\gamma$, target update frequency $n_{\text{target}}$, learning rate $\alpha$, offline sample ratio $\beta$, exploration rate $\epsilon$, action space $\mathcal{A}$.

1: Randomly initialize value function $f^\theta$.
2: Initialize target value function $\tilde{f} = f^\theta$.
3: Initialize online buffer $\mathcal{D} = \emptyset$.
4: Sample initial state $s \sim d_0$
5: **for** $t = 1, \ldots, T$ **do**
6:     Let $\pi$ be the $\epsilon$-greedy policy w.r.t. $f^\theta$ i.e., $\pi(s) = \text{argmax}_a f^\theta(s,a)$ with probability $1 - \epsilon$ and $\pi(s) = \mathcal{U}(\mathcal{A})$ with probability $\epsilon$.
    **// Online collection**
7:     Interact with the environment for one step:

$$a = \pi(s), s' \sim P(s,a), r \sim R(s,a).$$

8:     Update online buffer: $\mathcal{D} = \mathcal{D} \cup \{s,a,r,s'\}$.
    **// Discounted minibatch FQI using both online and offline data**
9:     **if** $t \mod n_{\text{value}} = 0$ **then**
10:       With probability $1 - \beta$: Sample a minibatch $D$ with size $n_{\text{minibatch}}$ from online buffer $\mathcal{D}$.
      Otherwise: Sample a minibatch $D$ with size $n_{\text{minibatch}}$ from offline buffer $\mathcal{D}^\nu$.
11:       Perform one-step gradient descent on $D$:

$$\theta = \theta - \alpha \nabla_\theta \hat{\mathbb{E}}_D \left( f^\theta(s,a) - r_i - \gamma \max_{a'} \tilde{f}(s',a') \right)^2.$$

12:     **end if**
    **// Delayed update of target function every** $n_{\text{target}}$ **updates**
13:     **if** $t \mod n_{\text{target}} = 0$ **then**
14:       Set target function to the current value function: $\tilde{f} = f^\theta$.
15:     **end if**
16:     Update $s \leftarrow s'$.
17: **end for**

---

### E.6 HYPERPARAMETERS

### E.6.1 COMBINATION LOCK

We provide the hyperparameters of Hy-Q in Table. 1. In addition, we provide the hyperparameters we tried for CQL baseline in Table. 2.

Table 1: Hyperparameters for Hy-Q in combination lock

|  | Value Considered | Final Value |
|---|---|---|
| Learning rate | {1e-2, 2e-2, 1e-3} | 2e-2 |
| Buffer size | {1e8} | 1e8 |
| Optimizer | {Adam, SGD} | Adam |
| Number of updates per iteration | {30, 300, 500} | 500 |
| Batch size | {512} | 512 |

### E.6.2 MONTEZUMA'S REVENGE

We provide the hyperparameter of Hy-Q in Table. 3. We reuse many hyperparameter choices from DQFD. Note that $[a, b]$ denotes a decreasing/increasing schedule from $a$ to $b$.

Table 2: Hyperparameters for CQL(DQN) in combination lock

|  | Value Considered | Final Value |
|---|---|---|
| Learning rate | {1e-3} | 1e-3 |
| Optimizer | {Adam} | Adam |
| Buffer size | {1e8} | 1e8 |
| Batch size | {512} | 512 |
| Discount Factor | {0.99} | 0.99 |
| Moving Average Factor $\tau$ | {0.01, 0.1, 1} | 0.01 |
| Weight on CQL loss $\alpha$ | {0, 0.1, 0.01} | 0.1 |

Table 3: Hyperparameter of Discounted Hy-Q in Montezuma's Revenge.

|  | Value Considered | Final Value |
|---|---|---|
| Learning rate | {6.25e-5, [1e-4,1e-5]} | [1e-4,1e-5] |
| Offline Schedule $\beta$ | {0.5,0.2,[0.2,0.01]} | [0.2,0.01] |
| Exploration $\epsilon$ rate | {[0.25,0.001]} | [0.25,0.001] |
| Minibatch size $n_{\text{minibatch}}$ | {32} | 32 |
| Weight decay (regularization) coefficient | {1e-5} | 1e-5 |
| Gradient Clipping | {10,20} | 10 |
| Discount factor $\gamma$ | {0.99} | 0.99 |
| Value function update frequency $n_{\text{update}}$ | {4} | 4 |
| Target function update frequency $n_{\text{target}}$ | {1000,2000,5000,10000} | 10000 |
| Buffer size | {$2^{20}$} | $2^{20}$ |
| PER Importance Sampling ratio | {[0.6,1]} | [0.6,1] |
| Online PER $\epsilon$ | {0.001} | 0.001 |
| Offline PER $\epsilon$ | {0.0001} | 0.0001 |
| Online PER Priority Coefficient | {0.4} | 0.4 |
| Offline PER Priority Coefficient | {1} | 1 |

