# OpenReview forum: "Hybrid RL: Using both offline and online data can make RL efficient"
_ICLR.cc/2023/Conference — ICLR 2023 poster_

### Official Review · Reviewer_bRLc · 2022-10-19

**Confidence:** 4
**Clarity, Quality, Novelty And Reproducibility:** See above
**Correctness:** 1
**Technical Novelty And Significance:** 2
**Empirical Novelty And Significance:** 3
**Recommendation:** 6

**Strength And Weaknesses:**

Strength: the theoretical analysis seems novel and the paper is easy to follow

Weaknesses:
1. The authors failed to properly place the algorithmic contribution in the literature. For Algorithm 1, I feel the idea is basically to fix the demonstration / offline data in experience replay and run some RL algorithm (e.g., FQI). This feeling is especially true when it is actually implemented with a discounted factor using a target network. This idea is not new, see, e.g., [x][y]. The authors failed to explicitly acknowledge the similarity between Algorithm 1 and existing works. The way to control catastrophic forgetting mentioned in Section 4 is also already done by [y] and the authors failed to acknowledge.
2.  As a result of (1), the main contribution is the theoretical analysis. So I must hold a higher standard when evaluating the contribution of the theoretical analysis. I, however, believe that Assumption 1 is way too strong. Can the author actually give a concrete meaningful example beyond the tabular setting where this assumption holds?
3. The selection of baselines is improper. The only baseline that is designed for both online and offline data is DQFD (Hester et al., 2018). However, it looks DQFD does imitation learning instead of RL for offline data. I believe using baselines methods that do RL for both online and offline data is more proper and more informative. In particular, the authors should compare with [x] and [y]. And **what is the fundamental difference between the algorithm that the authors actually implemented and [y]? The authors are using DQN and [y] uses DDPG. Other than this, I cannot see any major difference.** I feel the paper is quite far from making the following claim [z]
4. The domains for comparison can be much improved. For now the two domains are quite similar and requires guided exploration. I agree this is an important area of RL but considering only such areas is not enough to support the claim [z]. I think the authors should also compare in the settings where the reward is not so sparse. The authors should also consider offline dataset of different quality. D4RL seems to be a good complementary benchmarks. Otherwise the authors should consider tone down their claims.

[x] Lee, Seunghyun, et al. "Offline-to-online reinforcement learning via balanced replay and pessimistic q-ensemble." Conference on Robot Learning. PMLR, 2022.
[y] Vecerik, Mel, et al. "Leveraging demonstrations for deep reinforcement learning on robotics problems with sparse rewards." arXiv preprint arXiv:1707.08817 (2017).
[z] On the empirical side, several works consider combining offline expert demonstrations with online interaction (Rajeswaran et al., 2017; Hester et al., 2018; Nair et al., 2018; 2020; Vecerik et al., 2017). In our experiments, we find that these methods perform poorly when the offline dataset also contains experience from low-quality policies, while our method is robust to the quality of the offline data.

**Summary Of The Paper:**

This paper proposes a new method to run fitted Q learning on both online and offline data. Theoretical analysis is provided under several assumptions and empirical improvements over certain baselines are observed on certain tasks.

**Summary Of The Review:**

See above

==========================================
The revision cleared my major concern in credit assignment so I raised score accordingly.

---

> ### Author Response · Authors · 2022-11-10
> **Response to reviewer bRLc**
>
> ### General Response
>
> Thank you for you taking the time to review our paper. We'd like to point out that the most notable aspect of our work is that Hy-Q has both strong theoretical guarantees and empirical performance. Regarding theoretical results, our assumptions (completeness, coverage, bilinear rank) are widely used across RL theory, and in fact, our specific coverage condition is the weakest condition known to be tractable. We request the reviewer take another look at the theoretical aspect of our work in more detail. Regarding experiments, although the reviewer is right that other algorithms (like DDPG) may be adapted to the hybrid setting, none are known to have similar theoretical guarantees. Ours is the first work to provide an algorithm that is both *theoretically principled* and *practically useful*.
>
> ### Response to concern with Assumption 1
>
> Although Bellman completeness is somewhat strong, we note that: (i) it is widely used in RL theory, (ii) it, or some similarly strong condition, is provably necessary for offline RL, (iii) it holds far beyond the tabular setting, including in linear MDPs, linear quadratic regulators, the linear Bellman complete model (which generalizes both of these), state abstraction models, and Lipschitz continuous MDPs (Sun et al., 2019).
>
> Bellman completeness is one of the most standard assumptions used throughout the RL theory literature, starting from the classic paper in 1996 (Geoff Gordon, 1999), to the classic analysis of Fitted Q iteration (Remi et al., 2008), and more recent but well-known works on linear MDPs (Yang and Wang, 2019; Jin et al., 2020), and Bellman Eluder dimension (Jin et al., 2021). It is well known that algorithms like FQI and TD learning will diverge and fail, both in theory and in practice (see Foster et al., 2021 and Wang et al., 2021).
>
> We should also point out that, with function approximation, actor-critic algorithms like DDPG require (something like) completeness with respect to the $T^\pi$ operator for all policies $\pi$, which is a similarly strong (if not stronger) condition. Indeed, there are *no* known provably (computationally and statistically) efficient RL methods in the function approximation setting *whatsoever* without completeness or some related condition. Thus we believe it is unfair to penalize our paper for this assumption.
>
> Finally, our theorem can be easily relaxed to the setting where Bellman completeness only holds approximately, e.g., see Assumption 2 in Appendix A.
>
> ### Response to the other comments:
> > 1. I feel the idea is basically to fix the demonstration / offline data in experience replay and run some RL algorithm?
>
> We do not simply put offline data into one replay buffer. Instead, we maintain two buffers separately, and we carefully maintain the weight of the offline data in the least square regression objective through the entire learning process. This is demonstrated in the theory and in experiments. While this idea sounds simple, it does work in practice and in theory!
>
> > 2. The idea is not new.
>
> As we mentioned, our algorithm does sound simple, but it is provably correct and practically effective. Prior works lack strong theoretical guarantees and often fail to work well when offline data contains a non-trivial amount of low-quality transition data. We demonstrated this for DQFD in our experiments, and AWAC demonstrated itself in appendix (see section A.7 in https://openreview.net/pdf?id=OJiM1R3jAtZ).
>
> > 3. it looks DQFD does imitation learning instead of RL for offline data.
>
> We want to point out that this statement is not correct. The offline dataset of DQfD contains rewards, and DQfD does  Q-learning on the offline dataset (i.e., offline RL), with a supervised learning loss as a regularization.

---

> > ### Author Response · Authors · 2022-11-10
> > **Additional response**
> >
> > > 4. what is the fundamental difference between the algorithm that the authors actually implemented and [y]
> >
> > There are a few differences: (1) Hy-Q uses Q-learning instead of Actor-critic; (2) Hy-Q doesn't do n-step TD style update, since in off-policy case, without proper importance weighting, this could incur strong bias; While proper tuning on n could balance bias and variance, we do not need to tune such n-step at all in Hy-Q; (3) Instead of mixing offline data into a single replay buffer, Hy-Q maintains two replay buffers, and during learning (i.e., least square regression), offline data always has a non-trivial weight in the regression loss; (4) experimentally, [y] only considers expert demonstrations, we considered offline datasets with different amount of transitions from very low-quality policies, and showed Hy-Q is robust to low-quality transitions in offline data.  We will include a discussion on this list of differences in our revised version.
> >
> > While we agree that [y] sounds like a proper approach to combine offline data with online data in an actor-critic framework, [y] does not come with strong performance guarantees as we had, and [y] itself only focused on expert demonstrations. Indeed, we expect that to get provable guarantees for approach like [y], one needs stronger Bellman completeness conditions like the one proposed in Xie et al., 2021 (as we mentioned, without Bellman completeness, TD style update fails in the worst case). However, verifying the algorithm from [y] works for non-expert offline datasets theoretically and experimentally is beyond the scope of this work. We also agree that actor-critic can have better performance in some problems, but Q-learning style algorithm can have its advantage on other problems as well. Our work mainly focuses on studying Q-learning style update and thus we mainly focused on comparing to DQfD. The developing and analyzing Q-learning style approach in hybrid setting is a worthwhile contribution to the literature.
> >
> > > 5. The domains for comparison can be much improved.
> >
> > Our algorithm is based on Q learning, so it is not designed for continuous control benchmarks. We choose to focus on discrete action case, and we picked two environments -- one is popular in RL theory community where no deep RL baselines can solve it yet, the other one is a popular benchmark from deep RL community. Note that our algorithm works well on both benchmarks.
> > We also want to emphasize that the goal of our experiments is not to show that our method outperforms all existing empirical baselines on all benchmarks. Rather our experiments serve as a support to our theory results. We believe that the combination of experimental results and theoretical results have already demonstrated a strong statement on the advantage of hybrid RL over online/offline RL in both theory and in practice.
> > Regarding the dense reward comment, in this work we choose to focus on exploration heavy tasks since we want to show a clear advantage of hybrid RL over online RL. For dense reward, our theory still holds, and thus we believe that our algorithm will also work well in practice. We are happy to consider some dense reward setting for future work.
> >
> > > 6. The authors should also consider offline dataset of different quality.
> >
> > We want to point out that in our experiments, we already provide results on offline datasets with different qualities (easy/medium/hard). Please refer to our experiment section for more details.
> >
> > ### Regarding our claim in [z]:
> >
> > We revised our claim in a more rigorous manner in the updated version of the paper. Please refer to the last paragraph in Section 2. We meant that from our experiments, DQfD does not work well when offline data contains low-quality transitions, and AWAC itself demonstrated in the appendix that it is not robust to offline data containing low-quality transitions. However, we agree that an approach like [y] might still work when offline data contains bad transitions, though it is beyond the scope of this work to theoretically prove and empirically verify that (unfortunately [y] does not release code for us to quickly experiment it). Meanwhile, we are adapting [x]'s code to our experiment setup and we expect to report the result by the end of the discussion period.

---

> > > ### Author Response · Authors · 2022-11-10
> > > **References**
> > >
> > > ### References
> > >
> > > 1, Dylan J Foster, Akshay Krishnamurthy, David Simchi-Levi, and Yunzong Xu. Offline reinforcement learning: Fundamental barriers for value function approximation. In Conference on Learning Theory, 2021.
> > >
> > > 2, Tengyang Xie and Nan Jiang. Batch value-function approximation with only realizability. In International Conference on Machine Learning, 2021.
> > >
> > > 3, Wenhao Zhan, Baihe Huang, Audrey Huang, Nan Jiang, and Jason Lee. Offline reinforcement learning with realizability and single-policy concentrability. In Conference on Learning Theory, pp. 2730–2775. PMLR, 2022.
> > >
> > > 4, Jinglin Chen and Nan Jiang. Offline reinforcement learning under value and density-ratio realizability: the power of gaps. arXiv:2203.13935, 2022.
> > >
> > > 5, Tengyang Xie, Ching-An Cheng, Nan Jiang, Paul Mineiro, and Alekh Agarwal. Bellman-consistent pessimism for offline reinforcement learning. Advances in Neural Information Processing Systems, 2021a.
> > >
> > > 6, Lin Yang and Mengdi Wang. Sample-optimal parametric Q-learning using linearly additive features. In International Conference on Machine Learning, 2019.
> > >
> > > 7, Chi Jin, Zhuoran Yang, Zhaoran Wang, and Michael I Jordan. Provably efficient reinforcement learning with linear function approximation. In Conference on Learning Theory, 2020.
> > >
> > > 8, Chi Jin, Qinghua Liu, and Sobhan Miryoosefi. Bellman eluder dimension: New rich classes of RL problems, and sample-efficient algorithms. Advances in Neural Information Processing Systems, 2021a.
> > >
> > > 9, Wen Sun, Anirudh Vemula, Byron Boots, J. Andrew Bagnell. Provably Efficient Imitation Learning from Observation Alone. In International Conference on Machine Learning, 2019.
> > >
> > > 10, Remi Munos and Csaba Szepesvari. Finite-time bounds for fitted value iteration. Journal of Machine Learning Research, 2008.
> > >
> > > 11, Ruosong Wang, Dean P. Foster, Sham M. Kakade. What are the Statistical Limits of Offline RL with Linear Function Approximation? In International Conference on Learning Representations, 2021.
> > >
> > > 12, Geoffrey J. Gordon. Approximate Solutions to Markov Decision Processes. DOI:https://doi.org/10.1184/R1/6551972.v1, 1999.

---

### Official Review · Reviewer_mmKF · 2022-10-24

**Confidence:** 2
**Correctness:** 2
**Technical Novelty And Significance:** 3
**Empirical Novelty And Significance:** 3
**Recommendation:** 6

**Clarity, Quality, Novelty And Reproducibility:**

The writing of the paper is clear and the problem setting and algorithms are novel.

**Strength And Weaknesses:**

Strength: The paper considers a less studied setting and provides complete theoretical analysis of the proposed algorithm. The theoretical bound seems to have the optimal rate.

Weakness: The motivation of the hybrid RL setting is not very clear to me. In particular, the introduction part says that, to overcome the issue of sample inefficiency (of online RL), attention has turned to the offline RL setting. However, the information of each data point in online RL is surely much more than that of offline RL. Given the same sample size, we should always choose online RL over offline RL. The only setting that I can imagine that hybrid RL is useful is that there is already an offline dataset and we do not want to waste the data.

As for the theory part, similar confusion remains: the result in Theorem 1 seems similar to the convergence of offline RL, as it still requires the offline data set to have a good coverage condition number $C_{\pi}$. it is also not clear how the result is better than online RL (e.g., Du et al 2021). I am also wonder if using pessimism or optimism (from usual offline and online RL algorithms) in the hybrid RL algorithm is helpful to improve the sample efficiency. Or if the algorithm deliberately choose not to use them to reduce the computation cost.

**Summary Of The Paper:**

The paper considers hybrid reinforcement learning, which is a setting where both offline data and online data are available. The paper proposes a corresponding algorithm to be computation-efficient and sample-efficient under such a setting. The theoretical sample complexity upper bound shows both the characteristics of offline and online RL. The paper also provides experiments result to show the strength of the proposed algorithm.

**Summary Of The Review:**

The paper provides solid analysis of a novel RL algorithm in a less studied setting, in both theories and experiments. However, the contribution of the result seems not very clear, and thus I believe the paper is a little below the acceptance threshold. If the authors can well settle the concerns stated above, I may raise my score.

---

> ### Author Response · Authors · 2022-11-10
> **Response to reviewer mmKF**
>
> Thank you for reviewing our paper. We look forward to a productive Author-Reviewer discussion period. Below we provide our responses to the reviewer's questions.
>
> ### Response to Motivation in Practice
> > $\textbf{Question}$: What is the motivation of hybrid RL in practice?
>
> $\textbf{Answer}$: We are not sure if we fully understand your question. From what we understand, it seems that you are asking **"What is the motivation for hybrid RL in practice and why can not we just run online RL algorithms?"** We answer this question as follows:
>
> Our discussion in the introduction points to the fact that online RL often requires performing exploration which is challenging for many tasks. Furthermore, all theoretically rigorous online RL algorithms that have sample complexity guarantees (under our assumptions) are computationally inefficient.  Offline RL typically assumes one has an offline dataset that contains some transitions from a good policy (e.g., data from experts), which alleviates the burden from exploration. However, in general, the offline data may also have noisy trajectories (like samples collected via random policies) and may not be sufficient on its own to learn a good policy. Thus, the goal of this paper is to develop provably statistically and computationally efficient algorithms that can combine both offline and online data for learning. As the reviewer mentioned, hybrid RL indeed will be most valuable in the case where we have a ''good" offline dataset to start with. We mention that this setting has been widely studied in various prior empirical works (see references in the last section in section 2). However, ours is the first theoretically rigorous algorithm that also works well in practice. The experiments in the paper demonstrate the benefits of hybrid RL over online or offline RL. We hope this explains the motivation for this setting. Please let us know if this does not address your question, and we will be happy to follow up.
>
>
> ### Response to Motivation in Theory
> > $\textbf{Question}$: Is using pessimism or optimism (from usual offline and online RL algorithms) in the hybrid RL algorithm helpful to improve the sample efficiency?
>
> $\textbf{Answer}$: You are right, we deliberately choose to avoid optimism/pessimism, for the sake of computation efficiency (and statistical efficiency) -- see our detailed responses below.
>
> > $\textbf{Question}$: advantage of our result over online/offline RL alone
>
> $\textbf{Answer}$: $\textit{Comparison to offline RL methods:}$ offline RL methods rely on designing pessimism scheme for conservative policy optimization. Theoretical offline RL works are not computationally efficient due to the design of pessimism that operates over the whole version space, e.g., Xie et al., 2021, Chen and Jiang, 2022, Zhan et al., 2022. For deep offline RL baseline like CQL, we show that it does not work on M-revenge and Combination lock despite the fact that we tuned hyperparameters via online samples. We also would like to point out that our new definition of the transfer coefficient is the most general one in the literature on offline RL.
>
> $\textit{Comparison to online RL methods:}$ for the general function approximation setting that we consider here, existing provably efficient online methods are not computationally efficient. For instance, OLIVE (Jiang et al., 2017) and BLin-UCB (Du et al., 2021) are not even computationally efficient for tabular MDPs. They have never been implemented due to their known computation intractability.
> For a concrete example of provable benefits of Hybrid RL over online RL or offline RL, we would like to point the reviewer to Sec 5.1 again where for the linear Bellman complete model, Hy-Q has a polynomial running time, while no known online or offline RL algorithm can achieve that. This shows a clear separation between the hybrid setting and the online/offline setting.
>
> So in summary, operating under the hybrid RL setting allows us to avoid designing optimism and pessimism schemes, which often makes theoretical algorithms computationally intractable, and relies on good heuristics in practice. Our algorithm Hy-Q simply relies on least square regression oracles, which is a standard supervised learning technique.

---

> > ### Author Response · Authors · 2022-11-10
> > **References**
> >
> > ### References
> >
> > 1, Jinglin Chen and Nan Jiang. Offline reinforcement learning under value and density-ratio realizability: the power of gaps. arXiv:2203.13935, 2022.
> >
> > 2, Tengyang Xie, Ching-An Cheng, Nan Jiang, Paul Mineiro, and Alekh Agarwal. Bellman-consistent pessimism for offline reinforcement learning. Advances in Neural Information Processing Systems, 2021a.
> >
> > 3, Nan Jiang, Akshay Krishnamurthy, Alekh Agarwal, John Langford, and Robert E Schapire. Contextual decision processes with low Bellman rank are PAC-learnable. In International Conference on Machine Learning, 2017.
> >
> > 4, Simon Du, Sham Kakade, Jason Lee, Shachar Lovett, Gaurav Mahajan, Wen Sun, and Ruosong Wang. Bilinear classes: A structural framework for provable generalization in RL. In International Conference on Machine Learning, 2021.
> >
> > 5, Chi Jin, Qinghua Liu, and Sobhan Miryoosefi. Bellman eluder dimension: New rich classes of RL problems, and sample-efficient algorithms. Advances in Neural Information Processing Systems, 2021a.
> >
> > 6, Wenhao Zhan, Baihe Huang, Audrey Huang, Nan Jiang, and Jason Lee. Offline reinforcement learning with realizability and single-policy concentrability. In Conference on Learning Theory, pp. 2730–2775. PMLR, 2022.

---

### Official Review · Reviewer_WaHo · 2022-10-25

**Confidence:** 3
**Correctness:** 4
**Technical Novelty And Significance:** 3
**Empirical Novelty And Significance:** 3
**Recommendation:** 8

**Clarity, Quality, Novelty And Reproducibility:**

The paper is well-written and provides a non-trivial theoretical contribution to a hybrid RL.

**Strength And Weaknesses:**

[Strengths]
1. The paper presents a principled hybrid RL algorithm, which is supported by theoretical guarantees.
2. The algorithm is simple to implement and computationally efficient.
3. Experimental results show that the proposed Hy-Q outperforms online/offline baseline algorithms.


[Weaknesses]
1. It would be great to compare with more baseline algorithms that deal with both offline datasets and online experiences in the experiments, e.g. [1,2,3]
2. Page 4: There exists a gap between theory and practical implementation, i.e. using a fixed proportion of offline samples rather than forgetting them gradually, but I think this is minor.
3. Some notations are used without explicit definitions/explanations. What is $m_{off}$ and $m_{on}$ in Theorem 1? Also, they do not appear in the statement.


[Questions]
1. In the case of using flexible function approximators such as neural networks, is $C_\pi$ bounded?
2. Can Hy-Q be extended to continuous action spaces?
3. In Algorithm 1, the proportion of the offline dataset would decrease as the number of online samples increases. I am curious about the performance of this forgetting algorithm, rather than using a fixed proportion of offline samples.
4. I was wondering if the combination of Hy-Q and exploration strategy (e.g. RND) can improve or degrade the performance in the experiments.
5. The theoretical result was somewhat counter-intuitive to me. Hy-Q collects online samples via a 'greedy' policy w.r.t the learned value function; thus there is no exploration mechanism in the algorithm. Suppose an offline dataset was collected by a poor agent. Then, intuitively, exploitation alone w.r.t. the learned value function may be prone to get stuck in local optima and thus unable to learn a near-optimal policy. However, Corollary 1 is saying that Hy-Q can always learn a (near-)optimal policy with a sufficiently large number of samples. It would be great to provide some intuitive explanation about how 'greedy' data collection can always yield a near-optimal policy.


[1] Nair et al., AWAC: Accelerating Online Reinforcement Learning with Offline Datasets, 2020

[2] Lee et al., Offline-to-Online Reinforcement Learning via Balanced Replay and Pessimistic Q-Ensemble, 2021

[3] Xie et al., Policy Finetuning: Bridging Sample-Efficient Offline and Online Reinforcement Learning, 2022


**Summary Of The Paper:**

This paper presents Hybrid Q-Learning (Hy-Q), an algorithm for a hybrid RL setting, where the agent has both offline datasets and can interact with the environment in an online manner. Hy-Q learns a value function by fitted-Q iteration on both the offline dataset and the online experiences, where the online samples are collected by the greedy policy w.r.t. the learned value function. Theoretical analysis for the sample complexity of Hy-Q is provided. In addition, in the case of the linear Bellman completeness model, it can be shown that Hy-Q is a computationally efficient algorithm, which is in contrast to the existing methods that require solving NP-hard problems. In the experiments, Hy-Q outperforms both pure online and pure offline RL algorithms in Combination Lock and Montezuma's Revenge.


**Summary Of The Review:**

The paper presents a simple yet provably efficient algorithm for hybrid RL with theoretical guarantees, which I think is a novel and nice contribution. Still, in the experiments, comparisons with baseline algorithms that use both offline and online experiences are missing.
Lastly, one thing unclear to me was that Hy-Q uses a greedy policy for online data collection, but it can always learn a near-optimal policy without falling into some local optima, which is counterintuitive. I would like to raise my score if those concerns are addressed.
== post-rebuttal
Thanks for your clarifications, which addressed most of my concerns. I raise my score accordingly.

---

> ### Author Response · Authors · 2022-11-10
> **Response to reviewer WaHo**
>
> Thank you for reviewing our paper. We look forward to a productive Author-Reviewer discussion period. We want to emphasize that our main contributions are theoretical. In particular, we provide the first algorithm that provably solves various well-known and challenging theoretical RL problem settings computationally efficiently, e.g. linear Bellman complete, Section 5.1, etc. These settings did not have a provably computationally efficient algorithm prior to ours. At the same time, our provided algorithm is simple to implement, and works well in practice as shown by experiments on the challenging combination lock and M-revenge tasks. We will add comparison to one more baseline by the end of the discussion period according to reviewer's suggestion, and we hope the response below could address the other questions from the reviewers.
>
>
> ### Response to the weakness
> > $\textbf{Question}$: It would be great to compare with more baseline algorithms that deal with both offline datasets and online experiences in the experiments.
>
> $\textbf{Answer}$: We respond to each of the reviewer's suggestions below.
>
> [1] The AWAC method suffers performance degradation if the offline dataset has low quality (c.r. section A.7 in https://openreview.net/pdf?id=OJiM1R3jAtZ), which is exactly the setting we focus on and we believe no additional comparison with AWAC is required. Also AWAC is designed for low-dim continuous control, while in our experiment we focus on learning policy from high-dimensional images. We include DQfD as the baseline because it was developed for image-based tasks.
>
> [2] We agree this baseline is relevant. We are trying our best to perform comparison with it and update the result later, but now we want to highlight several difficulties in adapting this baseline to our experiments. a) The original code was designed for low-dimensional continuous control, which is not directly applicable to our discrete action and image-based environments. b) The proposed method heavily relies on CQL as a pre-training procedure to learn an ensemble of Q networks, but we have already demonstrated the CQL completely failed for M-revenge. c) The proposed method relies on training discriminators to learn density ratios between on-policy and offline data, which is a hard task especially when states are high-dimensional images. Note Hy-Q is a much simpler approach.
>
> [3] The algorithm is relevant but it is proposed for tabular MDP. Thus we do not believe it is applicable to our experiment settings.
>
> > $\textbf{Question}$: There exists a gap between theory and practical implementation.
>
> $\textbf{Answer}$: We want to clarify that the reviewer's confusion may rise from our statement from Thm.1 and Alg.1 in the old version. In our revised version, we specifically declares the number of online samples required in each iteration and the total number of offline sample (which depends on the number of iteration), reflecting the fact that we always maintain a significant portion of offline samples, both in theory and in practice.
>
> > $\textbf{Question}$: Some notations are used without explicit definitions/explanations. What is $m_{off}$ and $m_{on}$ in Theorem 1? Also, they do not appear in the statement.
>
> $\textbf{Answer}$: We will add the definition of $m_{off}$ (size of the total offline samples) and $m_{on}$ (size of the online samples at each iteration) in Thm.1 in the revised version. Note that $m_{off}$ and $m_{on}$ are input parameters to Algorithm 1, and do not appear in the sample complexity bound as they are themselves defined as functions of other problem parameters $(H, T, d)$. See the statement of Theorem 1 for their exact values.
>
> > $\textbf{Question}$: comparisons with baseline algorithms that use both offline and online experiences are missing.
>
> $\textbf{Answer}$: We include DQfD as a baseline that uses both offline and online data as one representative baseline for empirical hybrid RL baseline. Also we are adding another baseline as the reviewer suggests.

---

> > ### Author Response · Authors · 2022-11-10
> > **Response to the questions**
> >
> > > $\textbf{Question}$: In the case of using flexible function approximators such as neural networks, is $C_\pi$ bounded?
> >
> > $\textbf{Answer:}$ Yes! We want to point to one of the remarks we have in the paper (second paragraph above Thm.1 on page 5): ''Second, by using Bellman errors, both of these are bounded by notions involving density ratios (Kakade & Langford, 2002; Munos & Szepesvari, 2008; Chen & Jiang, 2019).'' That is saying, our coverage coefficient is very general and one way to interpret is that the coefficient is upper bounded by the density ratio which is independent of the value function approximation, including the neural network function approximation. To the best of our knowledge, our $C_{\pi}$ is the weakest coverage condition in the literature. The proposal of this definition is indeed one of our novel contributions.
> >
> > > $\textbf{Question}$: Can Hy-Q be extended to continuous action spaces?
> >
> > $\textbf{Answer:}$ Since Hy-Q is similar to Q-learning, it is not directly applicable to continuous control settings. We believe one very interesting future direction is to develop both theoretically and practically sound actor-critic style algorithms. Prior work like AWAC has taken an important step in this direction, though as we mentioned, its performance seems poor when offline data is low quality, and it lacks provably guarantees as well.
> >
> >
> > > $\textbf{Question}$: In Algorithm 1, the proportion of the offline dataset would decrease as the number of online samples increases. I am curious about the performance of this forgetting algorithm, rather than using a fixed proportion of offline samples.
> >
> > $\textbf{Answer:}$ We want to clarify that in our Algorithm 1, we design $m_{off}$ such that it is large enough so that in every least square regression loss, the offline data always has a non-trivial proportion. This avoids the forgetting issue, which allows us to prove strong theoretical results. Our implementation also relies on this intuition.
> > Furthermore, as is evident from the values of these variables in the statement of Theorem 1, (the total number of offline samples / total number of online samples) $\geq \Omega(1)$. This is also consistent with the experiments. We added clarifications in the revised version.
> >
> > > $\textbf{Question}$: I was wondering if the combination of Hy-Q and exploration strategy (e.g. RND) can improve or degrade the performance in the experiments.
> >
> > $\textbf{Answer:}$ This is a very interesting question and we think it may improve the performance of M-revenge. However, one great advantage of Hy-Q is that we do not need explicit exploration in the online phase. One reason is that designing exploration bonus is often tricky and often rely on good heuristics: we need to design different bonus for different tasks (e.g., RND does not work well beyond M-revenge, and count-based strategy does not work well on M-revenge). On the other hand, Hy-Q simply relies only on least square regression, which is a standard supervised learning problem.
> >
> >
> > > $\textbf{Question}$: It would be great to provide some intuitive explanation about how 'greedy' data collection can always yield a near-optimal policy.
> >
> > $\textbf{Answer:}$ We want to point out that Thm.1 is stating that we could find a $\epsilon$-optimal policy with respect to the *best policy* covered by the offline dataset. Thus your intuition is correct: if we have an offline dataset that does not cover any good policy (in terms of the transfer coefficient), then in theory we should not expect Hy-Q to return a good policy.
> >
> > We want to make one remark here: the quality of the offline data is controlled by $C_\pi$. If there is a high quality $\pi$ with bounded $C_\pi$, then Hy-Q will find it. In some sense, we rely on the fact that the offline data contains traces from a good policy. Note that this does not mean that one can treat the offline data as an expert dataset since it could additionally contain $(s,a,r,s')$ experiences from very bad policies (e.g., the hard dataset in our M-revenge experiment as the right-most plot in Fig.4).
> >
> > We also want to remark that the coefficient $C_\pi$ sometimes is a rather implicit metric (and this is why it is a novel and interesting construction): it is possible that the offline dataset is collected with low-performance policies, but the whole dataset actually contains some (s,a,r,s') tuples of which a good policy might visit as well (in the sense of how we define $C_\pi$). In that case, Hy-Q will still work well despite the quality of the policy used to collect the offline dataset.
> >
> > Finally, note that corollary 1 depends on $C_{\pi^\star}$, where $\pi^\ast$ is the optimal policy.

---

### Official Review · Reviewer_SJM5 · 2022-10-25

**Confidence:** 2
**Correctness:** 3
**Technical Novelty And Significance:** 2
**Empirical Novelty And Significance:** 2
**Recommendation:** 6

**Clarity, Quality, Novelty And Reproducibility:**



I think that the paper could better analyze  the reason of the different performance of the compared algorithms. For instance, If we analyze  DQFD  and Hy-Q, besides the difference in their losses, it seems that the main difference lies in how they sample offline and online data. I was wondering how DQFD would perform if it adopted the same scheme that the discounted Hy-Q algorithm follows to get batches from offline and online data (annealing beta), instead of forcefully maintaining offline data in its replay buffer. One could hypothesize that this might be the reason why DQFD consistently worsen its performance on Montezuma's revenge when deteriorating the quality of the offline dataset. Furthermore, I'm also wondering how a pretraining stage such as that proposed in DQFD would affect the discounted Hy-Q algorithm. I think that maybe these experiments would be beneficial to get a better grasp on how these two algorithms compare, and how these practical design decisions affect them.

**Strength And Weaknesses:**

Strengths
- The paper is well-written and the references appropriate.
- The proposed hybrid offline-online RL algorithm is simple and effective, and an extensive theoretical analysis of its properties is provided.

Weaknesses
- The comparison with other algorithms is weak. Just two simple problems are used, and the experiments could better analyze the reason of the different performance of the compared algorithms.


**Summary Of The Paper:**

This paper proposes a simple hybrid offline-online RL algorithm, and provides an extensive theoretical analysis to showcase its properties. The proposes algorithm is compared with baseline algorithms that fall into the categories of online, offline and hybrid RL, as well as "learning by imitation", in the "combination lock" and "Montezuma's revenge" problems.

**Summary Of The Review:**

The paper proposes a simple hybrid offline-online RL algorithm, it provides an extensive  analysis to showcase its properties and a comparison with other RL algorithms in two problems. I suggest to improve the experimental part and to provide more extensive experiments in more challenging problems, as well as a deeper analysis of the obtained results.

---

> ### Author Response · Authors · 2022-11-10
> **Response to reviewer SJM5**
>
> Thank you for reviewing our paper. We look forward to a productive Author-Reviewer discussion period. We want to emphasize that our main contributions are theoretical. In particular, we provide the first algorithm that provably solves various well-known and challenging theoretical RL problem settings computationally efficiently, e.g. linear Bellman complete, Section 5.1, etc. These settings did not have a provably computationally efficient algorithm prior to ours. At the same time, our provided algorithm is simple to implement, and works well in practice as shown by experiments on the challenging combination lock and M-revenge tasks.
>
> ### Response to the Questions
> > $\textbf{Question}$: how DQFD would perform if it adopted the same scheme that the discounted Hy-Q algorithm follows to get batches from offline and online data (annealing beta), instead of forcefully maintaining offline data in its replay buffer?
>
> $\textbf{Answer}$: To address the question, we performed new ablation experiments (see more details below). In summary, we tried the same scheme we had in Hy-Q for DQfD, but DQfD's result is still bad for hard offline datasets. We believe this is mostly because the DQfD algorithm incorporates supervised/imitation learning loss on the offline data, in both the pre-training and the RL fine-tuning phases. When offline data additionally contains transitions from low quality policies, this imitation learning style loss only degrades performance.
>
> > $\textbf{Question}$: Furthermore, I'm also wondering how a pretraining stage such as that proposed in DQFD would affect the discounted Hy-Q algorithm.
>
> $\textbf{Answer}$: The pre-training procedure in DQfD contains supervised learning loss, which we cannot use directly since that would deviate from our original algorithm. So we tried to use FQI/DQN on the offline data as warm start (the same as how we fit the offline data during online stage). This does not help the performance since FQI algorithm provably fails when the offline data does not have sufficient global coverage (i.e., our offline data is not diverse enough) [3]. This also demonstrates the benefit of using additional on-policy samples. Please see below for more experiment details.
>
>
> ### Response to the comment that the benchmarks in the experiment are simple
> We respectfully disagree with the statement that the benchmarks are simple. We specifically chose these domains for our experiments because they are very challenging and require non-trivial exploration to solve. The first environment, combination lock, is not solved by any deep RL algorithms yet (even the best exploration strategy in Deep RL, RND). The second benchmark, M-revenge, is also notoriously hard given the sparse reward and high-dimensional inputs. Prior deep RL works are specifically designed for solving this environment (e.g., [1,2]). As we also showed in our experiments, these two environments are hard for standard offline RL baseline as well (we used the representative CQL algorithm).
>
>
> ### New experiment results
>
> We perform two new experiments based on Reviewer SJM5's suggestions. The new experiments are in Appendix C (page 27) in the updated version.
> 1. We investigate the performance of DQfD if we use the same online-offline ratio as Hy-Q (section C.1). We replace DQfD's buffer selection scheme with Hy-Q's: the results show that this ablation does not significantly help DQfD. We believe that the major reason that prevents DQfD from making progress on the hard dataset is their supervised loss added to their offline loss.
> 2. We investigate the performance of Hy-Q if we add the offline FQI pretraining stage (section C.2). We show that adding the pretrain stage to Hy-Q does not significantly affect the performance of Hy-Q, and the result also suggests that offline FQI does not work on our problem setting, either.
>
>
> ### References
>
> [1] Burda, Yuri, et al. "Exploration by random network distillation." (2018).
>
> [2] Ecoffet, Adrien, et al. "Go-explore: a new approach for hard-exploration problems." (2019).
>
> [3] Remi Munos and Csaba Szepesvari. Finite-time bounds for fitted value iteration. Journal of Machine Learning Research, 2008.

---

### Public Comment · ~Zhiwei_Jia1 · 2022-11-11
**Related Work**

Hi authors, I really like your work on the Hybrid Q-Learning. Please consider citing the following paper that utilizes offline demos generated by distributed online agents for efficient online policy learning.

[1] Improving Policy Optimization with Generalist-Specialist Learning

ICML 2022

Zhiwei Jia, Xuanlin Li, Zhan Ling, Shuang Liu, Yiran Wu, Hao Su

---

> ### Author Response · Authors · 2022-11-20
> **Thank you for your reference**
>
> Hi,
>
> We appreciate your reference to the missing literature and your interest in our paper. We added the citation in our latest draft.
>
> Thanks,
> Paper1923 Authors

---

### Decision · Program_Chairs · 2023-01-20

**Decision:**

Accept: poster

**Justification For Why Not Higher Score:**

The algorithm is indeed simple, interesting, and effective. The reviewers raised several valid concerns regarding the novelty/experiments.

**Justification For Why Not Lower Score:**

The contribution is significant enough to be accepted.

**Metareview: Summary, Strengths And Weaknesses:**

Summary:

This paper proposes a simple hybrid offline-online RL algorithm that combines the data from both offline data and online exploration. It provides a theoretical analysis to showcase its efficiency. The paper also contains experimental results on some difficult Atari games and demonstrates the algorithm's effectiveness.

Strength:
- Well-written paper with both theoretical and empirical study
- The algorithm is simple and effective

Weakness:
- There exists a gap between theory and practical implementation
- The performance of the algorithm is upper bounded by the offline policy

Note: some recent works about pure online RL with general function approximations are missing.

**Note From Pc:**

if the above contains the word "oral" or "spotlight" please see: "oral" presentation means -> notable-top-5% and "spotlight" means -> notable-top-25%. As stated in our emails, we are disassociating presentation type from AC recommendations